# An Image is Worth More Than a Thousand Words: Towards Disentanglement in the Wild

**Aviv Gabbay**     **Niv Cohen**     **Yedid Hoshen**
School of Computer Science and Engineering
The Hebrew University of Jerusalem, Israel

Project webpage: http://www.vision.huji.ac.il/zerodim

## Abstract

Unsupervised disentanglement has been shown to be theoretically impossible without inductive biases on the models and the data. As an alternative approach, recent methods rely on limited supervision to disentangle the factors of variation and allow their identifiability. While annotating the true generative factors is only required for a limited number of observations, we argue that it is infeasible to enumerate all the factors of variation that describe a real-world image distribution. To this end, we propose a method for disentangling a set of factors which are only partially labeled, as well as separating the complementary set of residual factors that are never explicitly specified. Our success in this challenging setting, demonstrated on synthetic benchmarks, gives rise to leveraging off-the-shelf image descriptors to partially annotate a subset of attributes in real image domains (e.g. of human faces) with minimal manual effort. Specifically, we use a recent language-image embedding model (CLIP) to annotate a set of attributes of interest in a zero-shot manner and demonstrate state-of-the-art disentangled image manipulation results.

## 1   Introduction

High-dimensional data (e.g. images) is commonly assumed to be generated from a low-dimensional latent variable representing the true factors of variation [3, 29]. Learning to disentangle and identify these hidden factors given a set of observations is a cornerstone problem in machine learning, which has recently attracted much research interest [17, 22, 5, 30]. Recent progress in disentanglement has contributed to various downstream tasks as controllable image generation [44], image manipulation [14, 15, 42] and domain adaptation [33]. Furthermore, disentangled representations pave the way for better interpretability [18], abstract reasoning [40] and fairness [10].

A seminal study [29] proved that unsupervised disentanglement is fundamentally impossible without any form of inductive bias. While several different priors have been explored in recent works [24, 45], the prominent approach is to introduce a limited amount of supervision at training time, i.e. assuming that a few samples are labeled with the true factors of variation [30]. There are two major limitations of such semi-supervised methods; (i) Manual annotation can be painstaking even if it is only required for part of the images (e.g. 100 to 1000 samples). (ii) For real-world data, there is no complete set of semantic and interpretable attributes that describes an image precisely. For example, one might ask: *"Can an image of a human face be uniquely described with natural language?"*. The answer is clearly negative, as a set of attributes (e.g. age, gender, hair color) is far from uniquely defining a face.

Therefore, in this work we explore how to disentangle a few partially-labeled factors (named as *attributes of interest*) in the presence of additional completely unlabeled attributes. We then show that we can obtain labels for these attributes of interest with minimal human effort by specifying their optional values as adjectives in natural language (e.g. "blond hair" or "wearing glasses"). Specifically,

35th Conference on Neural Information Processing Systems (NeurIPS 2021).

we use CLIP [35], a recent language-image embedding model with which we annotate the training set images. As this model is already pretrained on a wide range of image domains, it provides rich labels for various visual concepts without any further manual effort in a zero-shot manner.

Nonetheless, leveraging general-purpose models as CLIP imposes a new challenge: among the attributes of interest, only part of the images are assigned to an accurate label. For this challenging disentanglement setting, we propose ZeroDIM, a novel method for Zero-shot Disentangled Image Manipulation. Our method disentangles a set of attributes which are only partially labeled, while also separating a complementary set of residual attributes that are never explicitly specified.

We show that current semi-supervised methods as Locatello et al. [30] perform poorly in the presence of residual attributes, while disentanglement methods that assume full supervision on the attributes of interest [14, 15] struggle when only partial labels are provided. First, we simulate the considered setting in a controlled environment with synthetic data, and present better disentanglement of both the attributes of interest and the residual attributes. Then, we show that our method can be effectively trained with partial labels obtained by CLIP to manipulate real-world images in high-resolution.

**Our contributions are summarized as follows:** (i) Introducing a novel disentanglement method for the setting where a subset of the attributes are partially annotated, and the rest are completely unlabeled. (ii) Replacing manual human annotation with partial labels obtained by a pretrained language-image embedding model (CLIP). (iii) State-of-the-art results on synthetic disentanglement benchmarks and real-world image manipulation tasks.

## 2   Related Work

**Semi-Supervised Disentanglement**  Locatello et al. [30] investigate the impact of a limited amount of supervision on disentanglement methods and observe that a small number of labeled examples is sufficient to perform model selection on state-of-the-art unsupervised models. Furthermore, they show the additional benefit of incorporating supervision into the training process itself. In their experimental protocol, they assume to observe all ground-truth generative factors but only for a very limited number of observations. On the other hand, methods that do not require labels for all the generative factors [7, 4, 11], rely on full-supervision for the observed ones. In a seminal paper, Kingma et al. [23] study the setting considered in our work, where some factors are labeled only in a few samples, and the other factors are completely unobserved. However, the approach proposed in [23] relies on exhaustive inference i.e. sampling all the possible factor assignments within the generative model. This exponential complexity inevitably limits the applicability of their approach for multi-attribute disentanglement. Nie et al. [31] propose a semi-supervised StyleGAN for disentanglement learning by combining the StyleGAN architecture with the InfoGAN loss terms. Although specifically designed for real high-resolution images, it does not natively generalize to unseen images. For this purpose, the authors propose an extension named Semi-StyleGAN-fine, utilizing an encoder of a locality-preserving architecture, which is shown to be restrictive in a recent disentanglement study [15]. Several other works [24, 45] suggest temporal priors for disentanglement, but can only be applied to sequential (video) data.

**Attribute Disentanglement for Image Manipulation**  The goal in this task is to edit a distinct visual attribute of a given image while leaving the rest of the attributes intact. Wu et al. [42] show that the latent space spanned by the style channels of StyleGAN [20, 21] has an inherent degree of disentanglement which allows for high quality image manipulation. TediGAN [43] and StyleCLIP [32] explore leveraging CLIP [35] in order to develop a text-based interface for StyleGAN image manipulation. Such methods that rely on a pretrained unconditional StyleGAN generator are mostly successful in manipulating highly-localized visual concepts (e.g. hair color), while the control of global concepts (e.g. age) seems to be coupled with the face identity. Moreover, they often require manual trial-and-error to balance disentanglement quality and manipulation significance. Other methods such as LORD [14] and OverLORD [15] allow to disentangle a set of labeled attributes from a complementary set of unlabeled attributes, which are not restricted to be localized (e.g. age editing). However, we show in the experimental section that the performance of LORD-based methods significantly degrades when the attributes of interest are only partially labeled.

**Joint Language-Image Representations**  Using natural language descriptions of images from the web as supervision is a promising direction for obtaining image datasets [6]. Removing the need for manual annotation opens the possibility of using very large datasets for better representation learning

which can later be used for transfer learning [19, 34, 39]. Radford et al. [36] propose learning a joint language-image representation with Contrastive Language-Image Pre-training (CLIP). The joint space in CLIP is learned such that the distance between the image and the text embedding is small for text-image pairs which are semantically related. This joint representation by CLIP was shown to have zero-shot classification capabilities using textual descriptions of the candidate categories. These capabilities were already used for many downstream tasks such as visual question answering [16], image clustering [9], image generation [37] and image manipulation [43, 32, 2].

## 3   Semi-Supervised Disentanglement with Residual Attributes

Assume a given set of images $x_1, x_2, ..., x_n \in \mathcal{X}$ in which every image $x_i$ is specified precisely by a set of true generative factors. We divide these factors (or attributes) into two categories:

**Attributes of Interest:** A set of semantic and interpretable attributes we aim to disentangle. We assume that we can obtain the labels of these attributes for a few training samples. We denote the assignment of these $k$ attributes of an image $x_i$ as $f_i^1, ..., f_i^k$.

**Residual attributes:** The attributes representing the remaining information needed to be specified to describe an image precisely. These attributes are represented in a single latent vector variable $r_i$.

As a motivational example, let us consider the task of disentangling human face attributes. The attributes of interest may include gender, age or hair color, while the residual attributes may represent the head pose and illumination conditions. Note that in real images, the residual attributes should also account for non-interpretable information e.g. details that relate to the facial identity of the person.

We assume that there exists an unknown function $G$ that maps $f_i^1, ..., f_i^k$ and $r_i$ to $x_i$:

$$x_i = G(f_i^1, ..., f_i^k, r_i) \tag{1}$$

A common assumption is that the true factors of variation are independent i.e. their density can be factorized as follows: $p(f^1, ..., f^k) = \prod_{j=1}^{k} p(f^j)$. Under this condition, the goal is to learn a representation that separates the factors of variation into independent components. Namely, a change in a single dimension of the representation should correspond to a change in a single generative factor. The representation of the residual attributes $r_i$, should be independent of all the attributes of interest. However, we only aim to learn a single unified representation for $r_i$, whose dimensions may remain entangled with respect to the residual attributes.

It should be noted that in real-world distributions such as real images, the true factors of variation are generally not independent. For example, the age of a person is correlated with hair color and the presence of facial hair. We stress that in such cases where the attributes are correlated, we restrict our attention to generating realistic manipulations: changing a target attribute with minimal perceptual changes to the rest of the attributes, while not learning statistically independent representations.

### 3.1   Disentanglement Model

Our model is aimed at disentangling the factors of variation for which at least *some* supervision is given. The provided labels are indicated by the function $\ell$:

$$\ell(i, j) = \begin{cases} 1 & f_i^j \text{ exists (attribute } j \text{ of image } i \text{ is labeled)} \\ 0 & \text{otherwise} \end{cases} \tag{2}$$

For simplicity, we assume that each of the attributes is a categorical variable and train $k$ classifiers (one per attribute) of the form $C^j : \mathcal{X} \to [m^j]$ where $m^j$ denotes the number of values of attribute $j$.

We optimize the classifiers using categorical cross-entropy loss, with the true labels when present:

$$\mathcal{L}_{cls} = \sum_{i=1}^{n} \sum_{j=1}^{k} \ell(i, j) \cdot H\left(\text{Softmax}\left(C^j(x_i)\right), f_i^j\right) \tag{3}$$

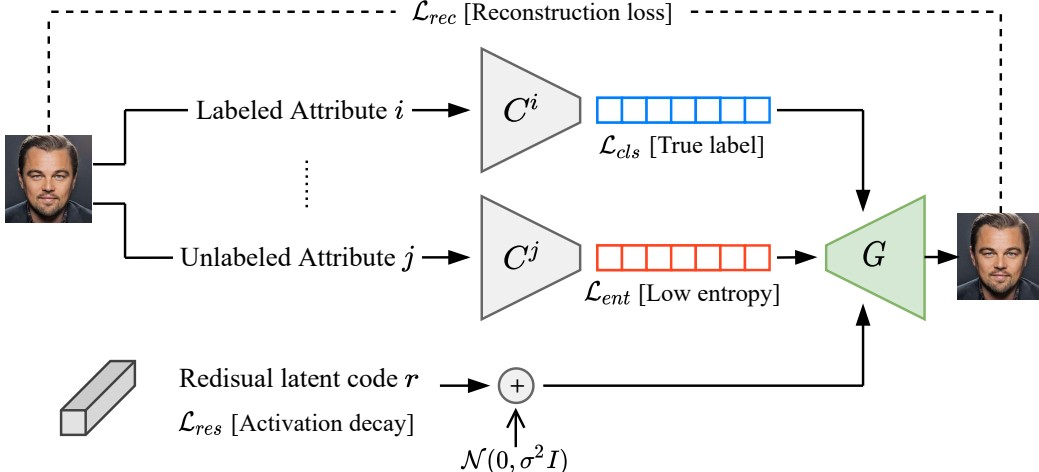

Figure 1: A sketch of our method. Given a partially-labeled dataset, the attribute classifiers $\{C^i\}$ are trained with two complementary objectives: i) Predicting the true labels for the labeled samples. ii) Predicting a low-entropy estimation of the unknown labels. The entropy term constrains the information capacity and prevents the leakage of information not related to the specific attribute, encouraging disentanglement. The latent code $r$ (one per image) is regularized and optimized to recover the minimal residual information that is required for reconstructing the input image.

where $H(\cdot, \cdot)$ denotes cross entropy. For samples in which the true labels of the attributes of interest are not given, we would like the prediction to convey the relevant information, while not leaking information on other attributes. Therefore, we employ an entropy penalty $\mathcal{L}_{ent}$ that encourages the attribute value of each sample to be close to any of the one-hot vectors describing the known values, limiting its expressivity. The penalty is set over the classifier prediction, using the entropy $H(\cdot)$:

$$\mathcal{L}_{ent} = \sum_{i=1}^{n} \sum_{j=1}^{k} (1 - \ell(i,j)) \cdot H\Big(\text{Softmax}\big(C^j(x_i)\big)\Big) \tag{4}$$

To train the downstream part of our model, we set the value for each of the attributes of interest $j$ according to the label if given, or our classifier prediction otherwise:

$$\tilde{f}_i^j = \begin{cases} f_i^j & \ell(i,j) = 1 \\ \text{Softmax}\big(C^j(x_i)\big) & \ell(i,j) = 0 \end{cases} \tag{5}$$

To restrict the information of the attributes of interest from "leaking" into the residual code, we constrain the amount of information in the residual code as well. Naively, all the image attributes, labeled and residual alike, might be encoded in the residual representations. As we aim for the residual representations to contain only the information not available in the attributes of interest, we regularize the optimized latent codes with Gaussian noise and an activation decay penalty [14]:

$$\mathcal{L}_{res} = \sum_{i=1}^{n} ||r_i||^2 \qquad r_i' = r_i + \mu, \mu \sim \mathcal{N}(0, I) \tag{6}$$

We finally employ a reconstruction loss to generate the target image:

$$\mathcal{L}_{rec} = \sum_{i=1}^{n} \phi(G(\tilde{f}_i^1, ..., \tilde{f}_i^k, r_i'), x_i) \tag{7}$$

where $\phi$ is a similarity measure between images, and is set to a mean-squared-error ($L_2$) loss for synthetic data and a perceptual loss for real images as suggested in [14].

Our disentanglement model is trained from scratch in an end-to-end manner, optimizing a generator $G$, classifiers $C^1, ..., C^k$ and a residual latent code $r_i$ per image $x_i$, with the following objective:

$$\min_{G,\{C^j\},\{r_i\}} \mathcal{L}_{disen} = \mathcal{L}_{rec} + \lambda_{cls}\mathcal{L}_{cls} + \lambda_{ent}\mathcal{L}_{ent} + \lambda_{res}\mathcal{L}_{res} \quad (8)$$

A sketch of our architecture is visualized in Fig. 1.

## 3.2 Implementation Details

**Latent Optimization**   We optimize over the latents codes $r_i$ directly as they are *not* parameterized by a feed-forward encoder. As discovered in [14], latent optimization improves disentanglement over encoder-based methods. The intuition is that at initialization time: each $r_i$ is initialized i.i.d, by latent optimization and therefore is totally uncorrelated with the attributes of interest. However, a feed-forward encoder starts with near perfect correlation (the attributes could be predicted even from the output of a randomly initialized encoder). At the end of training using latent optimization, we possess representations for the residual attributes for every image in the training set. In order to generalize to unseen images, we then train a feed-forward encoder $E_r$ to infer the residual attributes by minimizing: $\mathcal{L}_{enc} = \sum_{i=1}^{n} \|E_r(x_i) - r_i\|^2$.

**Warmup**   The attribute classifiers predict the true labels when present and a low-entropy estimation of the label otherwise, to constrain the information capacity. As the classifiers are initialized randomly, we activate the entropy penalty ($\mathcal{L}_{ent}$) only after a fixed number of epochs during training.

More implementation details are provided in the Appendix.

## 3.3 Experiments on Synthetic Datasets

We first simulate our disentanglement setting in a controlled environment with synthetic data. In each of the datasets, we define a subset of the factors of variation as attributes of interest and the remaining factors as the residual attributes (the specific attribute splits are provided in the Appendix). For each attribute of interest, we randomly select a specific number of labeled examples (100 or 1000), while not making use of any labels of the residual attributes. As a complementary experiment, we also show state-of-the-art results in the semi-supervised setting of disentanglement with no residual attributes, which is the setting studied in Locatello et al. [30] (see Appendix B).

We experiment with four disentanglement datasets whose true factors are known: Shapes3D [22], Cars3D [38], dSprites [17] and SmallNORB [27]. Note that partial supervision of 100 labels correspond to labeling 0.02% of Shapes3D, 0.5% of Cars3D, 0.01% of dSprites and 0.4% of SmallNORB.

### 3.3.1 Baselines

**Semi-supervised Disentanglement**   We compare against a semi-supervised variant of betaVAE suggested by Locatello et al. [30] which incorporates supervision in the form of a few labels for each factor. When comparing our method, we utilize the exact same betaVAE-based architecture with the same latent dimension ($d = 10$) to be inline with the disentanglement line of work. Our latent code is composed of two parts: a single dimension per attribute of interest (dimension $j$ is a projection of the output probability vector of classifier $C_j$), and the rest are devoted for the residual attributes $r_i$.

**Disentanglement of Labeled and Residual Attributes**   We also compare against LORD [14], the state-of-the-art method for disentangling a set of labeled attributes from a set of unlabeled residual attributes. As LORD assumes full-supervision on the labeled attributes, we modify it to regularize the latent dimensions of the partially-labeled attributes in unlabeled samples to better compete in our challenging setting. See Appendix for a discussion on the relation of our method to LORD.

Note that we do not compare to unsupervised disentanglement methods [17, 29] as they can not compete with methods that incorporate supervision, according to the key findings presented in [30].

Table 1: Evaluation on synthetic benchmarks using 1000 [or 100] labels per attribute of interest.

| | | D | C | I | SAP | MIG |
|---|---|---|---|---|---|---|
| **Shapes3D** | Locatello et al. [30] | 0.61 [0.03] | 0.61 [0.03] | 0.22 [0.22] | 0.05 [0.01] | 0.08 [0.02] |
| | LORD [14] | 0.60 [0.54] | 0.60 [0.54] | 0.58 [0.54] | 0.18 [0.15] | 0.43 [0.42] |
| | Ours | **1.00 [1.00]** | **1.00 [1.00]** | **1.00 [1.00]** | **0.30 [0.30]** | **1.00 [0.96]** |
| **Cars3D** | Locatello et al. [30] | 0.33 [0.11] | 0.41 [0.17] | 0.35 [0.22] | 0.14 [0.06] | 0.19 [0.04] |
| | LORD [14] | 0.50 [0.26] | 0.51 [0.26] | 0.49 [0.36] | 0.19 [0.13] | 0.41 [0.20] |
| | Ours | **0.80 [0.40]** | **0.80 [0.41]** | **0.78 [0.56]** | **0.33 [0.15]** | **0.61 [0.35]** |
| **dSprites** | Locatello et al. [30] | 0.01 [0.01] | 0.02 [0.01] | 0.13 [0.16] | 0.01 [0.01] | 0.01 [0.01] |
| | LORD [14] | 0.40 [0.16] | 0.40 [0.17] | 0.44 [0.43] | 0.06 [0.03] | 0.10 [0.06] |
| | Ours | **0.91 [0.75]** | **0.91 [0.75]** | **0.69 [0.68]** | **0.14 [0.13]** | **0.57 [0.48]** |
| **SmallNorb** | Locatello et al. [30] | 0.15 [0.02] | 0.15 [0.08] | 0.18 [0.18] | 0.02 [0.01] | 0.02 [0.01] |
| | LORD [14] | 0.03 [0.01] | 0.04 [0.03] | 0.30 [0.30] | 0.04 [0.01] | 0.04 [0.02] |
| | Ours | **0.63 [0.27]** | **0.65 [0.39]** | **0.53 [0.45]** | **0.20 [0.14]** | **0.40 [0.27]** |

### 3.3.2 Evaluation

We assess the learned representations of the attributes of interest using DCI [13] which measures three properties: (i) *Disentanglement* - the degree to which each variable (or dimension) captures at most one generative factor. (ii) *Completeness* - the degree to which each underlying factor is captured by a single variable (or dimension). (iii) *Informativeness* - the total amount of information that a representation captures about the underlying factors of variation. Tab. 1 summarizes the quantitative evaluation of our method and the baselines on the synthetic benchmarks using DCI and two other disentanglement metrics: SAP [26] and MIG [5]. It can be seen that our method learns significantly more disentangled representations for the attributes of interest compared to the baselines in both levels of supervision (100 and 1000 labels). Note that while other disentanglement metrics exist in the literature, prior work has found them to be substantially correlated [29].

Regarding the fully unlabeled residual attributes, we only require the learned representation to be informative of the residual attributes and disentangled from the attributes of interest. For evaluating these criteria, we train a set of linear classifiers, each of which attempts to predict a single attribute given the residual representations (using the available true labels). The representations learned by our method leak significantly less information regarding the attributes of interest. The entire evaluation protocol along with the quantitative results are provided in the Appendix for completeness.

### 3.4 Ablation Study

**Regularization Terms** We explore the contribution of the regularization terms introduced into our disentanglement objective (Eq. 8). Training our model without the entropy penalty $\mathcal{L}_{ent}$ results in inferior disentanglement of the attributes of interest, while removing the residual codes regularization $\mathcal{L}_{res}$ leads to a leakage of information related to the attributes of interest into the residual representations. The quantitative evidence from this ablation study is presented in the Appendix.

**Pseudo-labels** We consider a straightforward baseline in which we pretrain a classifier for each of the attributes of interest solely based on the provided few labels. We show in Tab. 2 that our method improves the attribute classification over these attribute-wise classifiers, implying the contribution of generative modeling to the discriminative-natured task of representation disentanglement [1]. Extended results from this study are provided in the Appendix.

Table 2: Average attribute classification accuracy using 1000 [or 100] labels per attribute.

| | Shapes3D | Cars3D | dSprites | SmallNORB |
|---|---|---|---|---|
| Pseudo-labels | 1.00 [0.84] | 0.82 [0.46] | 0.46 [0.28] | 0.51 [0.38] |
| Ours | **1.00 [0.99]** | **0.85 [0.51]** | **0.68 [0.41]** | **0.52 [0.39]** |

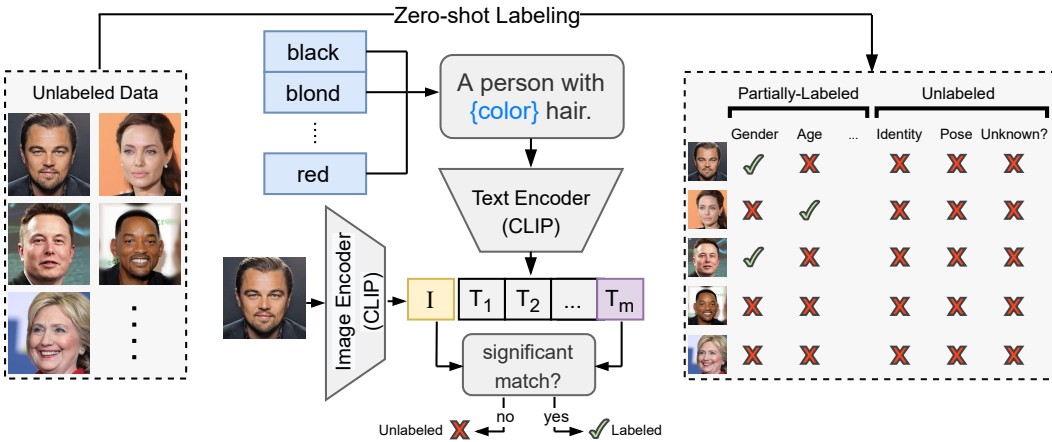

Figure 2: A visualization of our zero-shot labeling for real-world unlabeled data. We define a list of attributes of interest and specify their optional values as adjectives in natural language e.g. *blond* for describing *hair color*. To annotate a specific attribute, we embed the images along with several natural sentences containing the candidate labels (prompt engineering [35]) into the joint embedding space of a pretrained CLIP. The images which correlate the most with a query are assigned to a label. If none of the labels are assigned, the image remains unlabeled.

## 4 ZeroDIM: Zero-shot Disentangled Image Manipulation

In this section we make a further step towards disentanglement without manual annotation. We rely on the robustness of our method to partial-labeling which naturally fits the use of a general-purpose classification models such as CLIP [35]. CLIP is designed for zero-shot matching of visual concepts with textual queries. Our setting fits it in two aspects: (i) Only part of the attributes can be described with natural language. (ii) Only part of the images can be assigned to an accurate label.

### 4.1 Zero-shot Labeling with CLIP

Our use of CLIP [35] to provide annotations is driven by a short textual input. We provide short descriptions, suggesting a few possible values of each attribute of interest e.g. *hair color* can take one of the following: *"red hair", "blond hair"* etc. Yet, there are two major difficulties which prevent us from labeling all images in the dataset: (i) Even for a specified attribute, not necessarily all the different values (e.g. different hair colors) can be described explicitly. (ii) The classification capabilities of a pretrained CLIP model are limited. This can be expected, as many attributes might be ambiguous (*"a surprised expression" vs. "an excited expression"*) or semantically overlap (*"a person with glasses" or "a person with shades"*). To overcome the "noisiness" in our labels, we set a confidence criterion for annotation: for each value we annotate only the best $K$ matching images, as measured by the cosine distance of embedding pairs. Fig. 2 briefly summarizes this labeling process.

### 4.2 Experiments on Real Images

In order to experiment with real images in high resolution, we make two modifications to the method proposed in Sec. 3: (i) The generator architecture is adopted from StyleGAN2 [21], replacing the betaVAE decoder. (ii) An additional adversarial discriminator is trained with the rest of the modules for increased perceptual fidelity of the generated images, similarly to [15].

We demonstrate our zero-shot disentangled image manipulation on three different image domains: human faces (FFHQ [20]), animal faces (AFHQ [8]) and cars [25]. The entire list of attributes of interest used in each dataset can be found in the Appendix.

#### 4.2.1 Results

We compare our approach to two recent text-guided image manipulation techniques, TediGAN [43] and StyleCLIP [32], both utilizing a pretrained StyleGAN generator and a pretrained CLIP network.

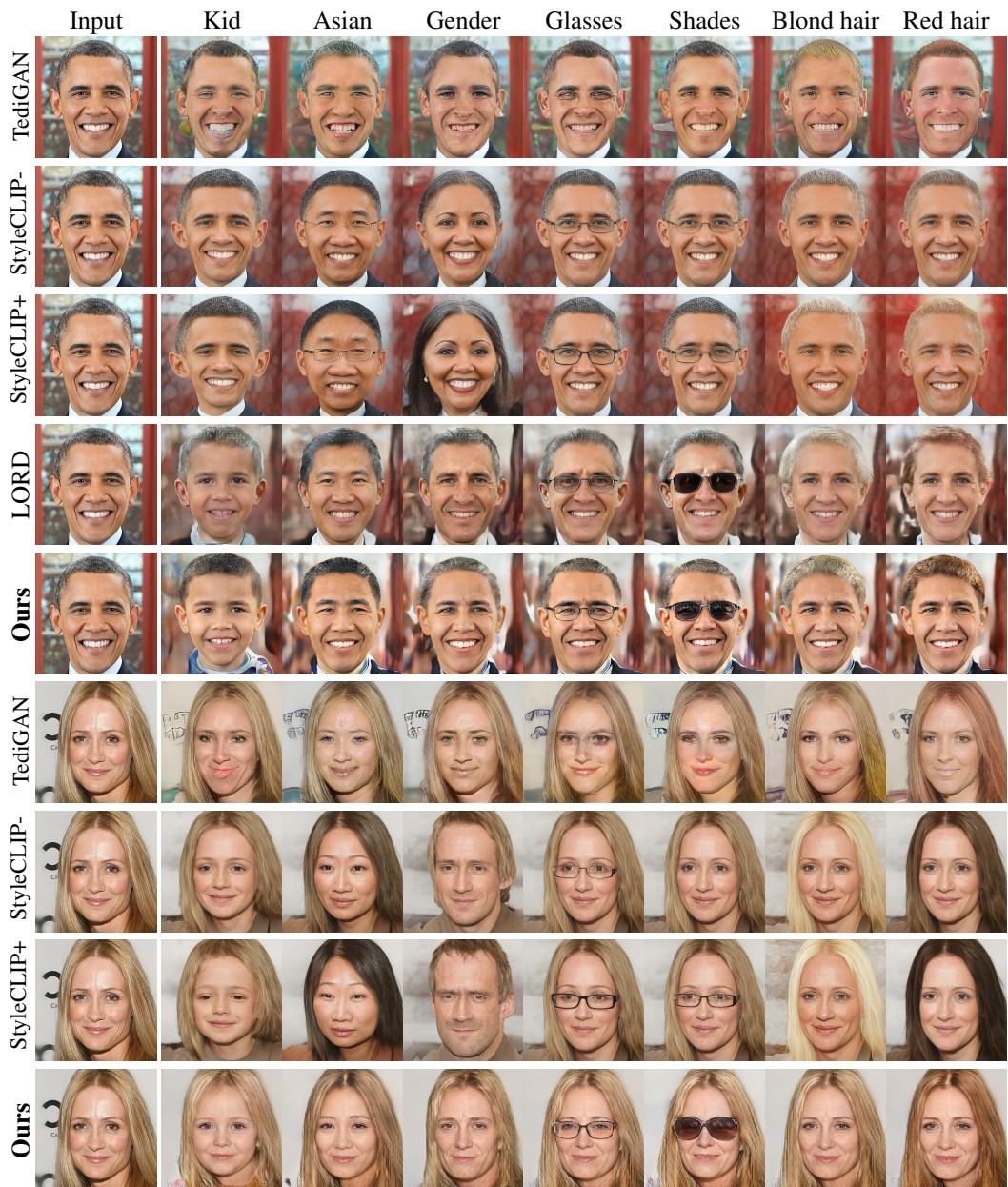

Figure 3: Zero-shot manipulation of human faces. StyleGAN-based approaches mainly disentangle highly-localized visual concepts (e.g. glasses) while global concepts (e.g. gender) seem to be entangled with identity. Moreover, their manipulation requires manual calibration, leading to negligible changes (e.g. invisible glasses) or extreme edits (e.g. translation to asian does not preserve identity). LORD does not require calibration but struggles to disentangle attributes which are not perfectly uncorrelated (e.g. the gender attribute is ignored and remains entangled with hair color). Our method generates highly disentangled results without manual tuning.

As can be seen in Fig. 3, despite their impressive image quality, methods that rely on a pretrained unconditional StyleGAN suffer from two critical drawbacks: (i) They disentangle mainly localized visual concepts (e.g. glasses and hair color), while the control of global concepts (e.g. gender) seems to be entangled with the face identity. (ii) The traversal in the latent space (the "strength" of the manipulation) is often tuned in a trial-and-error fashion for a given image and can not be easily calibrated across images, leading to unexpected results. For example, Fig. 3 shows two attempts to balance the manipulation strength in StyleCLIP ("-" denotes weaker manipulation and greater

| Input | Boerboel | Labradoodle | Chihuahua | Bombay Cat | Tiger | Lioness | Arctic Fox |
|-------|----------|-------------|-----------|------------|-------|---------|------------|

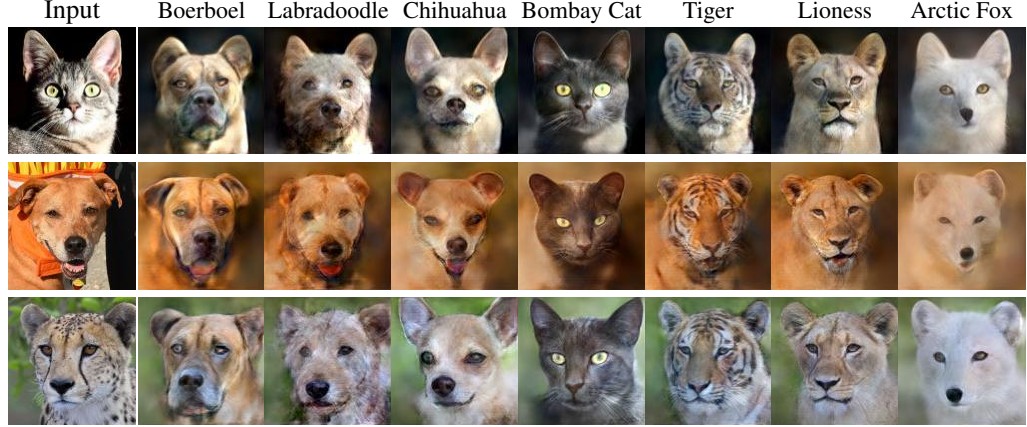

Figure 4: Zero-shot translation results of animal species on AFHQ. The pose of the animal (which is never explicitly specified) is preserved reliably while synthesizing images of different species.

| Input | Jeep | Sports | Family | Black | White | Red | Yellow |
|-------|------|--------|--------|-------|-------|-----|--------|

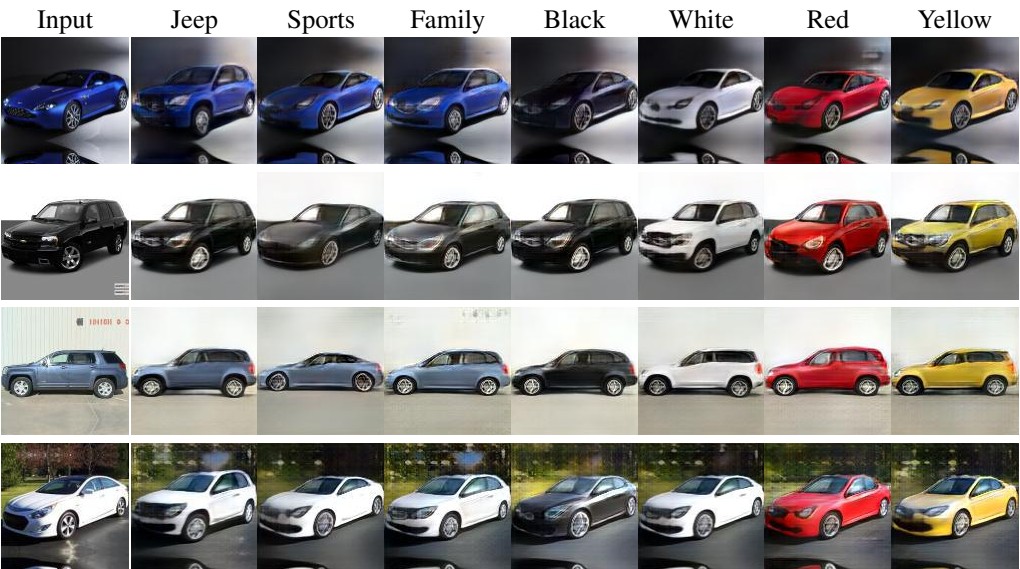

Figure 5: Zero-shot translation results of car types and colors.

disentanglement threshold than "+"), although both seem suboptimal. Our manipulations are highly disentangled and obtained without any manual tuning. Fig. 4 shows results of breed and species translation on AFHQ. The pose of the animal (which is never explicitly specified) is preserved reliably while synthesizing images of different species. Results of manipulating car types and colors are provided in Fig. 5. More qualitative and quantitative comparisons are provided in the Appendix.

### 4.3 Limitations

As shown in our experiments, our principled disentanglement approach contributes significantly to the control and manipulation of real image attributes in the absence of full supervision. Nonetheless, two main limitations should be noted: (i) Although our method does not require manual annotation, it is trained for a *fixed* set of attributes of interest, in contrast to methods as StyleCLIP which could adapt to a new visual concept at inference time. (ii) Unlike unconditional image generative models such as StyleGAN, reconstruction-based methods as ours struggle with synthesizing regions which exhibit large variability as hair-style. We believe that the trade-off between disentanglement and perceptual quality is an interesting research topic which is beyond the scope of this paper.

## 5   Conclusion

We studied a disentanglement setting, in which few labels are given only for a limited subset of the underlying factors of variation, that better fits the modeling of real image distributions. We then proposed a novel disentanglement method which is shown to learn better representations than semi-supervised disentanglement methods on several synthetic benchmarks. Our robustness to partial labeling enables the use of zero-shot classifiers which can annotate (only) a partial set of visual concepts. Finally, we demonstrated better disentangled attribute manipulation of real images. We expect the core ideas proposed in this paper to carry over to other modalities and applications.

## 6   Broader Impact

Disentanglement of images in real life settings bears great potential societal impact. On the positive side, better disentanglement may allow employing systems which are more invariant to protected attributes [28]. This may decrease the amount of discrimination which machine learning algorithms may exhibit when deployed on real life scenarios. On the negative side, along the possibility of malicious use of disentangled properties to discriminate intentionally; our work makes use of CLIP, a network pretrained on automatically collected images and labels. Such data are naturally prone to contain many biases. However, the technical contribution of our work can be easily adapted to future, less biased, pretrained networks.

The second aim of our work, namely, to produce disentangled zero shot image manipulations may present a broad impact as well. This is especially true when considering manipulation of human images. Disentangled image manipulation may assist synthetically creating more balanced datasets [12] in cases where the acquisition is extremely difficult (like rare disease). However, such methods may also be abused to create fake, misleading images [41]. On top of that, the manipulation methods themselves may introduce new biases. For example, our method is reliant of a finite set of values, which is far from describing correctly many attributes. Therefore, we stress that our method should be examined critically before use in tasks such as described here. We believe that dealing successfully with the raised issues in practice calls for future works, ranging from the technical sides, up to the regulatory and legislative ones.

**Acknowledgments**   We thank the anonymous reviewers for their thoughtful review and constructive feedback that helped clarifying the contribution and framing of this paper. We are grateful to Prof. Shmuel Peleg for the suggestions regarding the presentation of our work. This work was partly supported by the Federmann Cyber Security Research Center in conjunction with the Israel National Cyber Directorate. Computational resources were kindly supplied by Oracle Cloud Services.

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
