# Appendix - An Image is Worth More Than a Thousand Words: Towards Disentanglement in The Wild

## Table of Contents

## A   Semi-Supervised Disentanglement with Residual Attributes

### A.1   Implementation Details

Recall that our generative model gets as input the assignment of the attributes of interest together with the residual attributes and generates an image:

$$x = G(\tilde{f}^1, ..., \tilde{f}^k, r) \qquad (9)$$

where $\tilde{f}^j$ is the output probability vector (over $m^j$ different classes) of the classifier $C^j$. Generally, the embedding dimension of an attribute of interest $j$ within the generator is $d^j$. The embedding of each attribute of interest is obtained by the following projection: $\tilde{f}^j \cdot P^j$ where $P^j \in \mathbb{R}^{m^j \times d^j}$. The parameters of $P^j$ are optimized together with the rest of the parameters of the generator $G$. All the representations of the attributes of interest are concatenated along with the representation of the residual attributes before being fed into the generator $G$.

**Architecture for Synthetic Experiments**   To be inline with the disentanglement literature, we set $d^j = 1$ in the synthetic experiments, i.e. each attribute of interest should be represented in a single latent dimension. The entire latent code is therefore composed of $d = 10$ dimensions, $k$ of which are devoted for $k$ attributes of interest and $10 - k$ are for the residual attributes. The generator $G$ is an instance of the architecture of the betaVAE decoder, while each of the classifiers $C^j$ and the residual encoder $E_r$ is of the betaVAE encoder form. Detailed architectures are provided in Tab. 13 and Tab. 14 for completeness.

**Architecture for Experiments on Real Images**   In order to scale to real images in high resolution, we replace the betaVAE architecture with the generator architecture of StyleGAN2, with two modifications; (i) The latent code is adapted to our formulation and forms a concatenation of the representations of the attributes of interest and the residual attributes. (ii) We do not apply any noise injection, unlike the unconditional training of StyleGAN2. Note that in these experiments we train an

Table 3: Attribute splits for the synthetic benchmarks.

| Dataset | Attributes of Interest | Residual Attributes |
|---|---|---|
| Shapes3D | floor color, wall color, object color | scale, shape, azimuth |
| Cars3D | elevation, azimuth | object |
| dSprites | scale, x, y | orientation, shape |
| SmallNORB | elevation, azimuth, lighting | category, instance |

additional adversarial discriminator to increase the perceptual quality of the synthesized images. A brief summary of the architectures is presented in Tab. 15 and 16 for completeness. The architecture of the feed-forward residual encoder trained in the second stage is influenced by StarGAN-v2 [8] and presented in Tab. 17.

**Optimization** All the modules of our disentanglement model are trained from scratch, including the generator $G$, classifiers $C^1, ..., C^k$ and a residual latent code $r_i$ per image $x_i$. While in the synthetic experiments each of the $k$ attributes of interest is embedded into a single dimension, we set the dimension of each attribute to $8$ and of the residuals to $256$ in the experiments on real images. We set the learning rate of the latent codes to $0.01$, of the generator to $0.001$ and of the attribute classifiers to $0.0001$. The learning rate of the additional discriminator (only trained for real images) is set to $0.0001$. The practice of using a higher learning rate for the latent codes is motivated by the fact that the latent codes (one per image) are updated only once in an entire epoch, while the parameters of the other modules are updated in each mini-batch. For each mini-batch, we update the parameters of the models and the relevant latent codes with a single gradient step each. The different loss weights are set to $\lambda_{cls} = 0.001, \lambda_{ent} = 0.001, \lambda_{res} = 0.0001$. In order to stabilize the training, we sample both supervised and unsupervised samples in each mini-batch i.e. half of the images in each mini-batch are labeled with at least one true attribute of interest.

## A.2  Evaluation Protocol

We assess the learned representations of the attributes of interest using DCI [13] which measures three properties: (i) *Disentanglement* - the degree to which each variable (or dimension) captures at most one generative factor. (ii) *Completeness* - the degree to which each underlying factor is captured by a single variable (or dimension). (iii) *Informativeness* - the total amount of information that a representation captures about the underlying factors of variation. Tab. 1 summarizes the quantitative evaluation of our method and the baselines on the synthetic benchmarks using DCI and two other disentanglement metrics: SAP [26] and MIG [5].

Regarding the fully unlabeled residual attributes, we only require the learned representation to be informative of the residual attributes and disentangled from the attributes of interest. As we cannot expect the codes here to be disentangled from one another, we cannot use the standard disentanglement metrics (e.g. DCI). For evaluating these criteria, we train a set of linear classifiers, each of which attempts to predict a single attribute given the residual representations (using the available true labels). The results and comparisons can be found in Sec. A.4.

The specific attribute splits used in our synthetic experiments are provided in Tab. 3.

## A.3  Baseline Models

### A.3.1  Synthetic Experiments

**Semi-Supervised betaVAE (Locatello et al. [30]):** We compare with the original implementation[1] of the semi-supervised variant of betaVAE provided by Locatello et al. [30]. Note that we consider the *permuted-labels* configuration in which the attribute values are not assumed to exhibit a semantic order. While the order of the values can be exploited as an inductive bias to disentanglement, many attributes in the real world (e.g. human gender or animal specie) are not ordered in a meaningful way.

**LORD (Gabbay and Hoshen [14]):** As this method assumes full-supervision on the attributes of interest, we make an effort to adapt to the proposed setting and regularize the latent codes of the

---

[1]https://github.com/google-research/disentanglement_lib

unlabeled images with activation decay penalty and Gaussian noise. We do it in a similar way to the regularization applied to the residual latent codes. While other forms of regularization might be considered, we believe it is the most trivial extension of LORD to support partially-labeled attributes.

### A.3.2   Real Images Experiments

**LORD (Gabbay and Hoshen [14]):** We use the same method as in the synthetic experiments, but adapt it to real images. Similarly to the adaptation applied for our method, we replace the betaVAE architecture with StyleGAN2, add adversarial discriminator, and increase the latent codes dimensions. The limited supervision here is supplied by CLIP [35] as in our method.

**StyleCLIP (Patashnik et al. [32]):** We used the official repository[2] of the authors. The parameters were optimized to obtain both visually pleasing and disentangled results. Two configurations were chosen, one for the "StyleCLIP+" comparison ($\alpha = 4.1$), and one for "StyleCLIP-" ($\alpha = 2.1$). We always used $\beta = 0.1$.

**TediGAN (Xia et al. [43]):** To optimize the trade-off between visually pleasing results and applying the desired manipulation, the value of "$clip\_loss$" $= 5.0$ was chosen. Other parameters were given their default value in the official supplied code[3].

We note that to obtain the results presented in StyleCLIP and TediGAN papers, the authors adjusted these parameters per image. While this might be a reasonable practice for an artist utilizing these methods, it is not part of our setting.

### A.3.3   Relation to LORD [14]

Recall that we aim to achieve disentanglement in the absence of full supervision on the attributes of interest (i.e. the supervised class in [14]). We stress and show that methods as LORD [14] that indeed aim to disentangle the attributes of interest from a unified set of residual attributes, only work when full supervision is available on the attributes of interest and struggle when only partial labels are provided (see Tab. 1 and Fig. 6,7,8). More specifically, our method can be seen as an extension of LORD [14] for cases where the attributes of interest are observed only in a very few samples. Our method can therefore also leverage off-the-shelf zero-shot image classifiers such as CLIP, in order to be applied without the need to manually annotate even a small set of images. The few labels obtained with CLIP can not be used effectively by LORD [14], as demonstrated in our experiments.

From a technical perspective, there are two fundamental differences between our method and LORD [14]: (i) LORD is a fully latent-based model i.e. no classifiers are trained with the generator in the first stage. The latent codes of the attributes of interest are optimized directly (and shared between all instances with the same label). Here we provide a hybrid latent-amortized approach where attribute codes are learned in a latent fashion, similarly to LORD, but they are weighted using the probabilities emitted by an amortized classifier. (ii) We introduce an additional term $\mathcal{L}_{ent}$ which enables our method to perform well when very limited supervision exists for the attributes of interest.

### A.4   Additional Results

### A.4.1   Synthetic Experiments

**Disentanglement of the Residual Code**   We report the accuracy of linear classifiers in predicting the values of the different attributes from the residual code in Tab. 7. Ideally, the residual code should contain all the information about the residual attributes, and no information about the attributes of interest. Therefore, for each dataset we expect the attributes of interest (first row of each dataset, colored in red) to be predicted with low accuracy. The residual attributes (second row, in green) are the ones that *should* be encoded by the residual code.

We see that our method almost always provides better disentanglement (worse prediction) between the residual code and the attributes of interest. Although the residual code in the method by Locatello et al. [30] sometimes provide better predictions regarding the residual attributes, it comes at the expense of containing a lot information regarding the attributes of interest. Keeping in mind that our

---

[2]https://github.com/orpatashnik/StyleCLIP
[3]https://github.com/IIGROUP/TediGAN

Table 4: Ablation for $\mathcal{L}_{ent}$ using 1000 [or 100] labels per attribute of interest.

|  |  | D | C | I | SAP | MIG |
|---|---|---|---|---|---|---|
| Shapes3D | Ours w/o $\mathcal{L}_{ent}$ | 0.99 [0.99] | 0.98 [0.98] | 0.98 [0.98] | 0.28 [0.26] | 0.94 [0.91] |
|  | Ours | **1.00 [1.00]** | **1.00 [1.00]** | **1.00 [1.00]** | **0.30 [0.30]** | **1.00 [0.96]** |
| Cars3D | Ours w/o $\mathcal{L}_{ent}$ | 0.74 [0.39] | 0.74 [0.40] | 0.71 [0.43] | 0.22 [0.11] | 0.57 [0.34] |
|  | Ours | **0.80 [0.40]** | **0.80 [0.41]** | **0.78 [0.56]** | **0.33 [0.15]** | **0.61 [0.35]** |

Table 5: Ablation for $\mathcal{L}_{res}$ using 1000 labels per attribute of interest (lower is better).

|  |  | floor color | wall color | object color |
|---|---|---|---|---|
|  | Ours w/o $\mathcal{L}_{res}$ | 0.23 | 0.28 | 0.18 |
| Shapes3D | Ours | **0.11** | **0.12** | **0.14** |
|  | Random Chance (optimal) | 0.10 | 0.10 | 0.10 |

goal is to disentangle the attributes of interest between themselves, and from the residual code, we conclude the our suggested method performs better on this metric as well.

**Regularization Terms** We provide an ablation study of the different terms in Eq. 8. We first show in Tab. 4 that without the entropy penalty $\mathcal{L}_{ent}$ we obtain inferior disentanglement of the attributes of interest. For evluating the importance of the residual codes regularization $\mathcal{L}_{res}$ we measure the accuracy of classifying the attributes of interest from the residual representations using logistic regression. The results in Tab. 5 highlight the contribution of this term.

**Pseudo-labels** We show in Tab. 8 a full version of the average ablation table of attribute classification supplied on the main text (Tab. 2). The results suggest that the attribute classification accuracy is improved by our method, compared to using the same architecture as a classifier trained only on the labeled samples.

### A.4.2 Real Images Experiments

**Quantitative Evaluation** The quantitative evaluation of disentanglement in real images is challenging as no ground truth annotations are available for all the attributes and the attributes are not completely independent. For evaluation purposes, the paper includes quantitative metrics on synthetic benchmarks and many qualitative comparisons on real images. We further consider quantitative metrics for evaluation of our method on real images of human faces. We assess the performance by Attribute-Dependency (AD) (proposed in [42]): we measure the degree to which manipulation of a certain attribute induces changes in other attributes, as measured by classifiers for these attributes. We rely on 40 pretrained classifiers for attributes in CelebA, in order to cope with real images, where the exact factors of variation are not observed. Intuitively, disentangled manipulations should induce smaller changes in other attributes (lower AD is better). In addition, we report the *manipulation strength* for each attribute, as measured by the normalized change to the logit of the classifier of the target attribute. Note that the manipulation strength can be negative in cases where the manipulation causes an opposite effect to the attribute.

Tab. 6 shows the AD scores and manipulation strength of all methods while manipulating different attributes of interest. We stress that quantitative measurements of this sort are not perfect and can sometimes be misleading. However, let us briefly review the main trends reflected by these metrics (the same trends are clearly visualized in Fig. 6,7,8): (i) StyleCLIP tends to over manipulate the desired attribute and causes changes to other attributes of the input image, resulting in inferior disentanglement and leading to higher AD scores. This can be clearly seen when changing gender. (ii) LORD struggles to disentangle attributes which are not perfectly uncorrelated e.g. manipulating gender does not affect the input image at all (low manipulation strength which results in a misleading low AD score) while adding beard to females leads to gender swapping (higher AD scores). Note that TediGAN mostly introduces artifacts without manipulating the desired attribute (low manipulation strengths), and therefore maintains misleading low AD scores. The manipulation strength of the ethnicity attribute is not reported due to the lack of a pretrained classifier, and the manipulation

Table 6: Evaluation of disentanglement measured by Attribute Dependency (↓) and [manipulation strength (↑)], on real human face images.

| | Age | Beard | Ethnicity | Gender | Glasses | Hair Color |
|---|---|---|---|---|---|---|
| TediGAN [43] | 0.39 [0.04] | 0.38 [-] | 0.41 [-] | 0.40 [0.02] | 0.31 [0.18] | 0.37 [0.28] |
| StyleCLIP [32] | 0.45 [-0.07] | 0.42 [-] | 0.40 [-] | 0.78 [0.57] | 0.35 [0.22] | 0.44 [0.17] |
| LORD [14] | 0.41 [0.13] | 0.65 [-] | 0.38 [-] | 0.36 [0.04] | 0.46 [0.19] | 0.38 [0.26] |
| Ours | 0.40 [0.12] | 0.36 [-] | 0.40 [-] | 0.44 [0.20] | 0.49 [0.23] | 0.37 [0.28] |

strength of beard is not assessed as this attribute is correlated with other attributes (e.g. gender) and the manipulation should not cause any effect in many cases.

**Qualitative Visualizations**    We provide more qualitative results on FFHQ (Fig. 6,7,8) along with a comparison to TediGAN, StyleCLIP and LORD. More qualitative results on AFHQ and Cars are shown in Fig. 9 and Fig. 10.

### A.5   Training Resources

Training our models on the synthetic datasets takes approximately $3 - 5$ hours on a single NVIDIA RTX 2080 TI. Training our model on the largest real image dataset (FFHQ) at $256 \times 256$ resolution takes approximately $4$ days using two NVIDIA V100 GPU.

## B   Semi-Supervised Disentanglement without Residual Attributes

We evaluate our method on the synthetic datasets with all the factors of variation treated as attributes of interest, holding out no residual factors at all. This is the setting studied by Locatello et al. [30]. While this is not the task we aim to solve in this paper, our method performs better than [30], using the same beta-VAE based architecture, as can be seen in Tab. 9. This highlights the advantage of latent optimization for disentanglement as discussed in [14].

## C   Zero-shot Labeling with CLIP

### C.1   Implementation Details

We will provide here a comprehensive description of our method to annotate given images according to the attributes supplied by the user, utilizing the CLIP [35] network. For the annotation, the user provides a list of *attributes*, and for each attribute, a list of possible values indexed by $w \in [m^j]$: $s_j^w$ (see Sec. C.2). For each attribute $j$ and every possible value index $w$, we infer its embedding $u_j^w \in \mathbb{R}^{512}$ using language embedding head of the CLIP model $\phi_{lang}$:

$$u_j^w = \phi_{lang}(s_j^w) \tag{10}$$

To obtain a similar embedding for images, we pass each image $x_1, x_2, ..., x_n \in \mathcal{X}$ through the vision-transformer (ViT) head of the clip model $\phi_{vis}$. We obtain the representation of each image in the joint embedding space $v_i \in \mathbb{R}^{512}$:

$$v_i = \phi_{vis}(x_i) \tag{11}$$

We are now set to assign for each image $i$ and each attribute $j$ their assignment value $a_{ij}$ (one of the $[m^j]$ possible values, or alternatively, the value "$-1$"). For each image we set the value $w \in [m^j]$ to $a_{ij}$, if the image embedding $v_i$ is among the top $K$ similar images to the value embedding $u_j^w$ in the cosine similarity metric (noted by $d$). With a slight abuse of notation:

$$a_{ij} = \{w \mid \left|\{l \mid d(u_j^w, v_l) \leq d(u_j^w, v_i)\}\right| \leq K\} \tag{12}$$

where $|\cdot|$ is the number of elements in a set.

The value "$-1$" is assigned to $a_{ij}$ in cases where our zero shot classification deem that image uncertain: if an image not among the top $K$ matches for any of the values $u_j^w$, or if it is among the top $K$ matches for more than one of them.

Table 7: Accuracy of factor predictions from the residual code on the synthetic benchmarks, using 1000 [or 100] labels per attribute of interest. We indicate beside each attribute its number of values. Lower accuracy in predicting the attributes of interest and higher accuracy in predicting the residual attributes indicate better disentanglement.

**Dataset: Shapes3D  Attributes: floor, wall, object  Residuals: scale, shape, azimuth**

| | floor color [10] | wall color [10] | object color [10] |
|---|---|---|---|
| Locatello et al. [30] | 1.00 [1.00] | 1.00 [1.00] | 0.87 [1.00] |
| LORD [14] | 0.96 [0.80] | 0.87 [0.75] | 0.35 [0.49] |
| Ours | **0.11 [0.13]** | **0.12 [0.15]** | **0.14 [0.15]** |

| | scale [8] | shape [4] | azimuth [15] |
|---|---|---|---|
| Locatello et al. [30] | 0.15 [0.34] | 0.29 [0.32] | 0.59 [0.77] |
| LORD [14] | 0.25 [0.17] | 0.38 [0.37] | 0.20 [0.15] |
| Ours | **0.75 [0.47]** | **0.97 [0.79]** | **0.79 [0.48]** |

**Dataset: Cars3D  Attributes: elevation, azimuth  Residuals: object**

| | elevation [4] | azimuth [24] |
|---|---|---|
| Locatello et al. [30] | 0.44 [0.46] | 0.86 [0.88] |
| LORD [14] | 0.35 [0.36] | 0.43 [0.55] |
| Ours | **0.29 [0.33]** | **0.28 [0.26]** |

| | object [183] |
|---|---|
| Locatello et al. [30] | **0.64 [0.44]** |
| LORD [14] | 0.37 [0.27] |
| Ours | 0.51 [0.23] |

**Dataset: dSprites  Attributes: scale, x, y  Residuals: orientation, shape**

| | scale [6] | x [32] | y [32] |
|---|---|---|---|
| Locatello et al. [30] | 0.38 [0.35] | 0.24 [0.20] | 0.18 [0.34] |
| LORD [14] | 0.33 [0.31] | 0.20 [0.15] | 0.26 [0.18] |
| Ours | **0.20 [0.20]** | **0.04 [0.05]** | **0.04 [0.04]** |

| | orientation [40] | shape [3] |
|---|---|---|
| Locatello et al. [30] | 0.04 [0.03] | **0.44 [0.44]** |
| LORD [14] | 0.03 [0.03] | 0.43 [0.42] |
| Ours | **0.06 [0.06]** | 0.40 [0.41] |

**Dataset: SmallNORB  Attributes: elevation, azimuth, lighting  Residuals: category, instance**

| | elevation [9] | azimuth [18] | lighting [6] |
|---|---|---|---|
| Locatello et al. [30] | **0.16 [0.17]** | 0.12 [**0.10**] | 0.91 [0.91] |
| LORD [14] | **0.16 [0.17]** | 0.11 [0.12] | 0.89 [0.87] |
| Ours | **0.16** [0.18] | **0.10 [0.10]** | **0.24 [0.24]** |

| | category [5] | instance [10] |
|---|---|---|
| Locatello et al. [30] | 0.48 [**0.54**] | 0.14 [**0.14**] |
| LORD [14] | 0.47 [0.50] | 0.14 [**0.14**] |
| Ours | **0.59** [0.41] | **0.16** [0.14] |

A high value of $K$ indicates that many images will be labeled for each value $u_j^w$, resulting in a more extensive, but sometime noisy supervision. Many images are not mapped in a close proximity to the embedding of all the sentences describing them, or are not described by any of the user supplied values. Therefore, we would not like to use very large values for $K$. A low value of $K$ sets a more limited supervision, but with more accurate labels. The value of $K$ chosen in practice is indicated in the tables in Sec. C.2.

Table 8: Attribute classification accuracy using 1000 [or 100] labels per attribute.

| Dataset | Attribute [No. of Values] | Pseudo-labels | Ours |
|---|---|---|---|
| Shapes3D | floor color [10] | **1.00** [0.92] | **1.00** [**1.00**] |
| | wall color [10] | **1.00** [0.91] | **1.00** [**1.00**] |
| | object color [10] | **1.00** [0.68] | **1.00** [**0.98**] |
| Cars3D | elevation [4] | 0.81 [0.51] | **0.85** [**0.52**] |
| | azimuth [24] | 0.83 [0.40] | **0.85** [**0.49**] |
| dSprites | scale [6] | 0.45 [0.36] | **0.51** [**0.47**] |
| | x [32] | 0.46 [0.23] | **0.76** [**0.33**] |
| | y [32] | 0.46 [0.25] | **0.76** [**0.42**] |
| SmallNORB | elevation [9] | **0.28** [**0.19**] | **0.28** [0.16] |
| | azimuth [18] | 0.34 [**0.11**] | **0.36** [**0.11**] |
| | lighting [6] | **0.92** [0.86] | **0.92** [**0.90**] |

Table 9: Evaluation on synthetic benchmarks in the setting where there are no residual attributes [30], using 1000 labels per attribute of interest (mean [std]).

| | | DCI Disentanglement | SAP | MIG |
|---|---|---|---|---|
| Shapes3D | Locatello [30] | 0.99 [0.001] | 0.23 [0.01] | 0.75 [0.05] |
| | Ours | **1.00** [0.001] | **0.37** [0.001] | **0.99** [0.01] |
| Cars3D | Locatello [30] | 0.58 [0.05] | 0.14 [0.01] | 0.25 [0.01] |
| | Ours | **0.59** [0.06] | **0.19** [0.01] | **0.49** [0.01] |
| dSprites | Locatello [30] | 0.46 [0.03] | 0.07 [0.001] | 0.33 [0.01] |
| | Ours | **0.62** [0.01] | **0.09** [0.01] | **0.39** [0.01] |
| SmallNORB | Locatello [30] | 0.43 [0.02] | 0.13 [0.01] | 0.24 [0.01] |
| | Ours | **0.68** [0.01] | **0.31** [0.22] | **0.52** [0.01] |

## C.2 Attribute Tables

We include the entire list of the attributes and their possible values for each of the datasets annotated with CLIP. These values are used by our method, and by the LORD [14] baseline. The lists were obtained as follows. **FFHQ**: We aggregated human face descriptors, similar to the ones used by other methods, to obtain the attributes in Tab. 10. **AFHQ**: To obtain candidate values for animal species we used an online field guide. We randomly selected 200 images of the AFHQ dataset and identified them to obtain the list in Tab. 11. **Cars**: We used a few cars types and colors as described in Tab. 12.

For comparison with competing methods we used the attribute descriptions brought in their cited paper, or in the code published by the authors. If no similar attribute appeared in the competing methods, we tried a few short descriptions of the attribute, similar to the ones used by our method.

## D Datasets

**FFHQ [20]** $70,000$ high-quality images containing considerable variation in terms of age, ethnicity and image background. We use the images at $256 \times 256$ resolution. We follow [21] and use all the images for training. The images used for the qualitative visualizations contain random images from the web and samples from CelebA-HQ.

**AFHQ [8]** $15,000$ high quality images categorized into three domains: cat, dog and wildlife. We use the images at $128 \times 128$ resolution, holding out 500 images from each domain for testing.

**Cars [25]** $16,185$ images of 196 classes of cars. The data is split into $8,144$ training images and $8,041$ testing images. We crop the images according to supplied bounding boxes, and resize the images to $128 \times 128$ resolution (using "border reflect" to avoid distorting the image due to aspect ratio changes).

# E  Extended Qualitative Visualizations

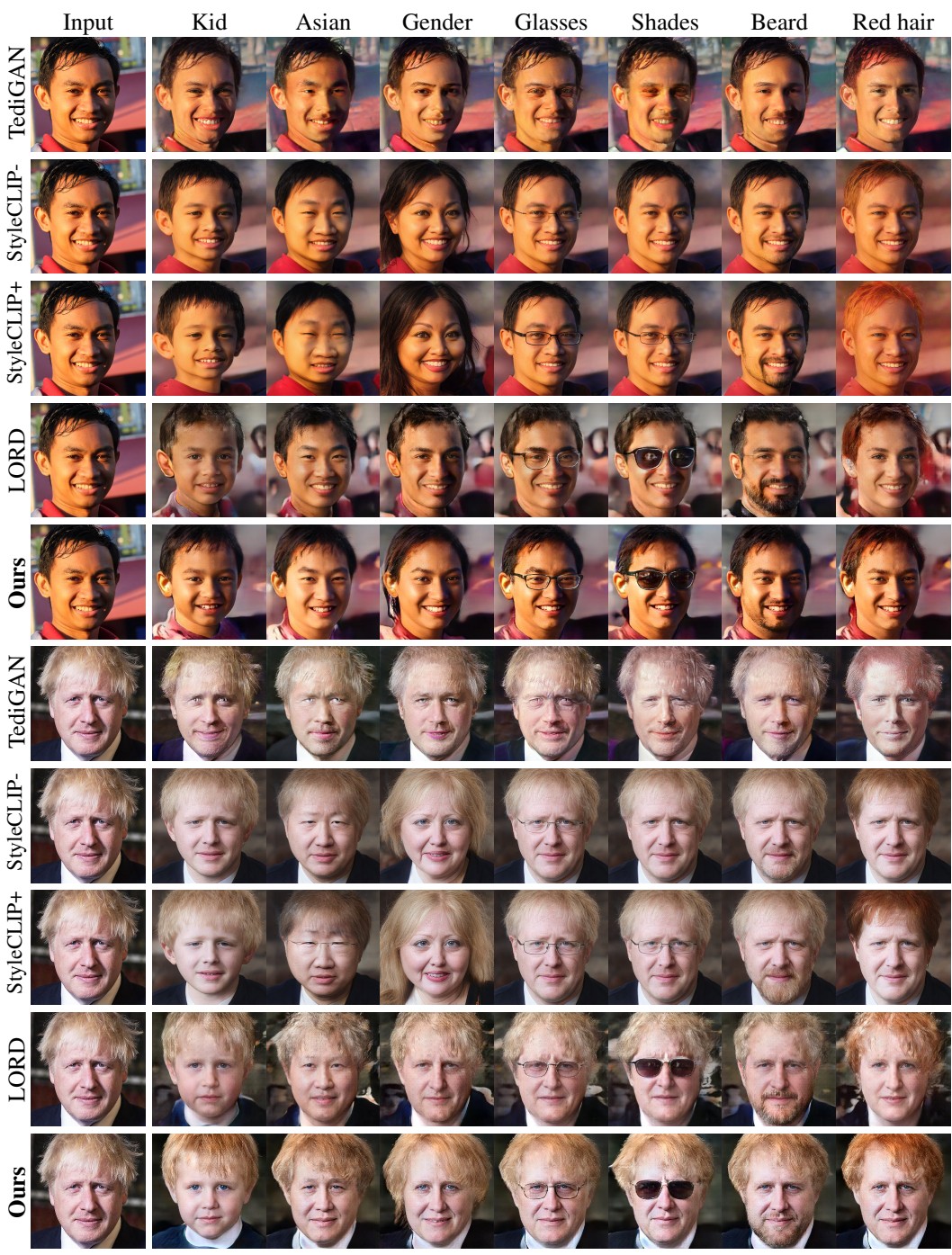

Figure 6: Zero-shot manipulation of human faces. StyleGAN-based approaches (TediGAN and StyleCLIP) mainly disentangle highly-localized visual concepts (e.g. beard) while global concepts (e.g. gender) seem to be entangled with identity and expression. Moreover, their manipulation requires manual calibration, leading to negligible changes (e.g. invisible glasses) or extreme edits (e.g. translation to asian does not preserve identity). LORD does not require calibration but struggles to disentangle attributes which are not perfectly uncorrelated (e.g. the gender attribute is ignored and remains entangled with beard and hair color). Our method generates highly disentangled results without manual tuning.

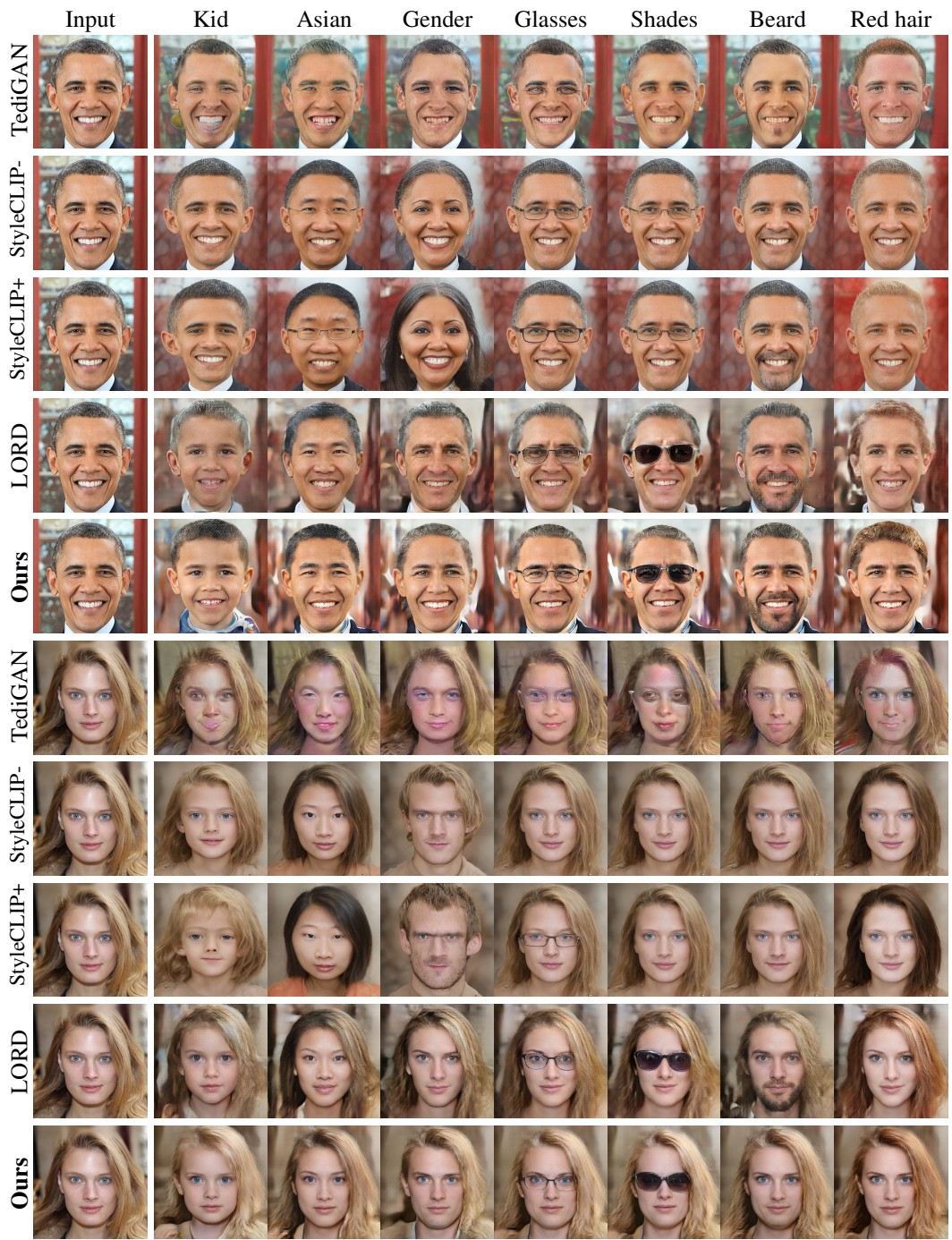

Figure 7: Zero-shot manipulation of human faces. StyleGAN-based approaches (TediGAN and StyleCLIP) mainly disentangle highly-localized visual concepts (e.g. beard) while global concepts (e.g. gender) seem to be entangled with identity and expression. Moreover, their manipulation requires manual calibration, leading to negligible changes (e.g. invisible glasses) or extreme edits (e.g. translation to asian does not preserve identity). LORD does not require calibration but struggles to disentangle attributes which are not perfectly uncorrelated (e.g. the gender attribute is ignored and remains entangled with beard and hair color). Our method generates highly disentangled results without manual tuning. Note that all manipulations are subject to attribute correlation e.g. beard is not significantly added to females.

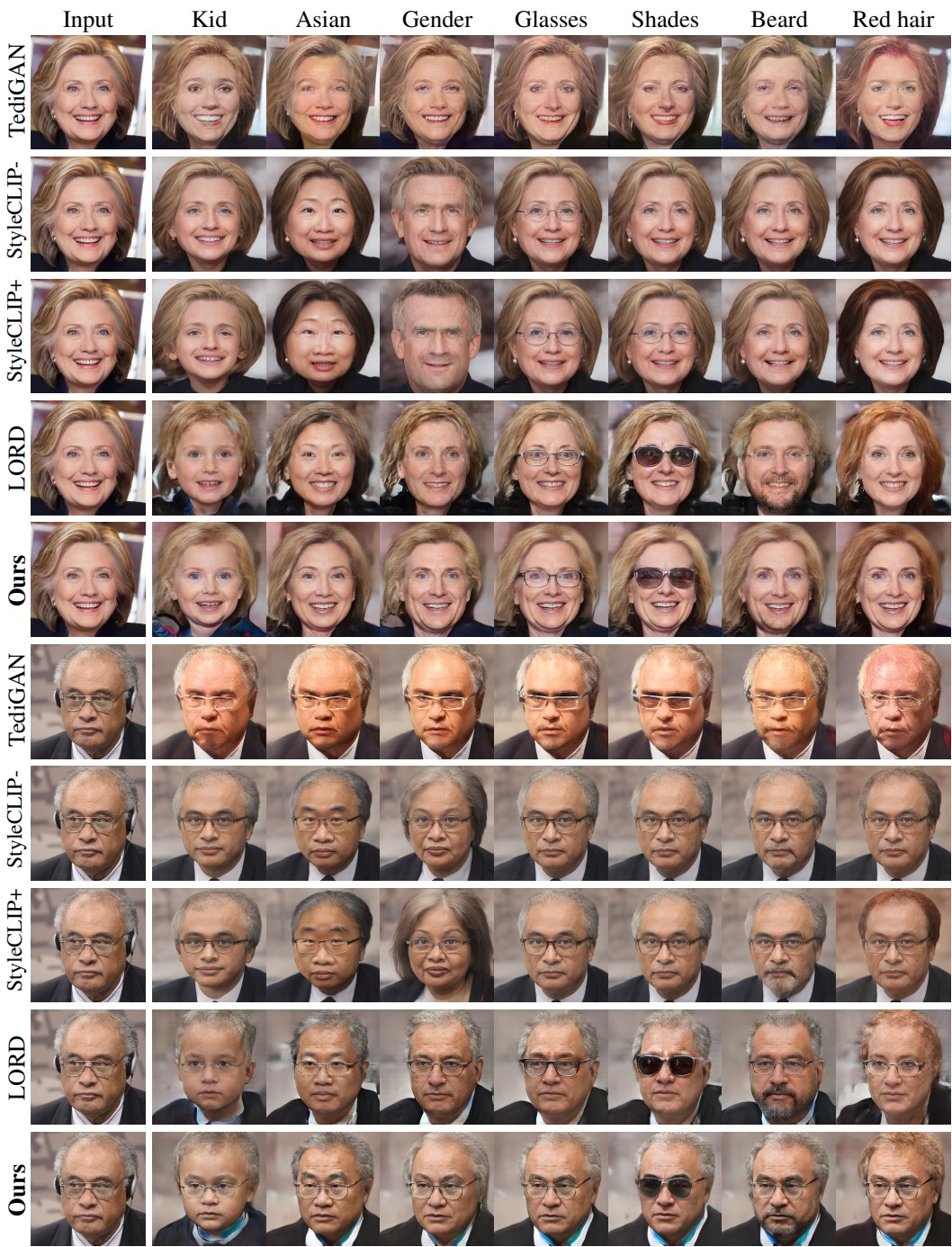

Figure 8: Zero-shot manipulation of human faces. StyleGAN-based approaches (TediGAN and StyleCLIP) mainly disentangle highly-localized visual concepts (e.g. beard) while global concepts (e.g. gender) seem to be entangled with identity and expression. Moreover, their manipulation requires manual calibration, leading to negligible changes (e.g. invisible glasses) or extreme edits (e.g. translation to asian does not preserve identity). LORD does not require calibration but struggles to disentangle attributes which are not perfectly uncorrelated (e.g. the gender attribute is ignored and remains entangled with beard and hair color). Our method generates highly disentangled results without manual tuning. Note that all manipulations are subject to attribute correlation e.g. beard is not significantly added to females.

Input  Boerboel  Labradoodle  Husky  Chihuahua  Cheetah  Jaguar  Bombay cat

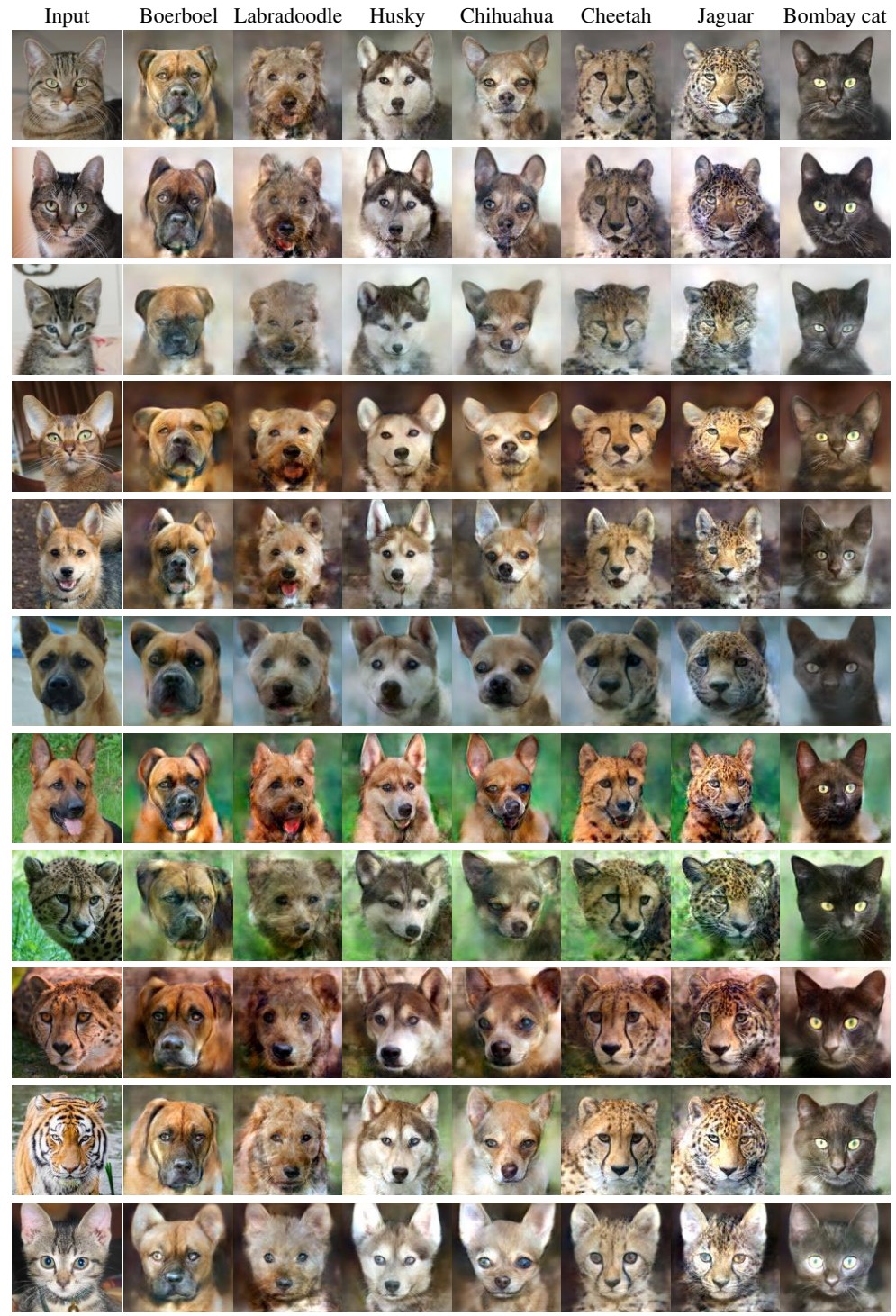

Figure 9: More zero-shot translation results of animal species on AFHQ.

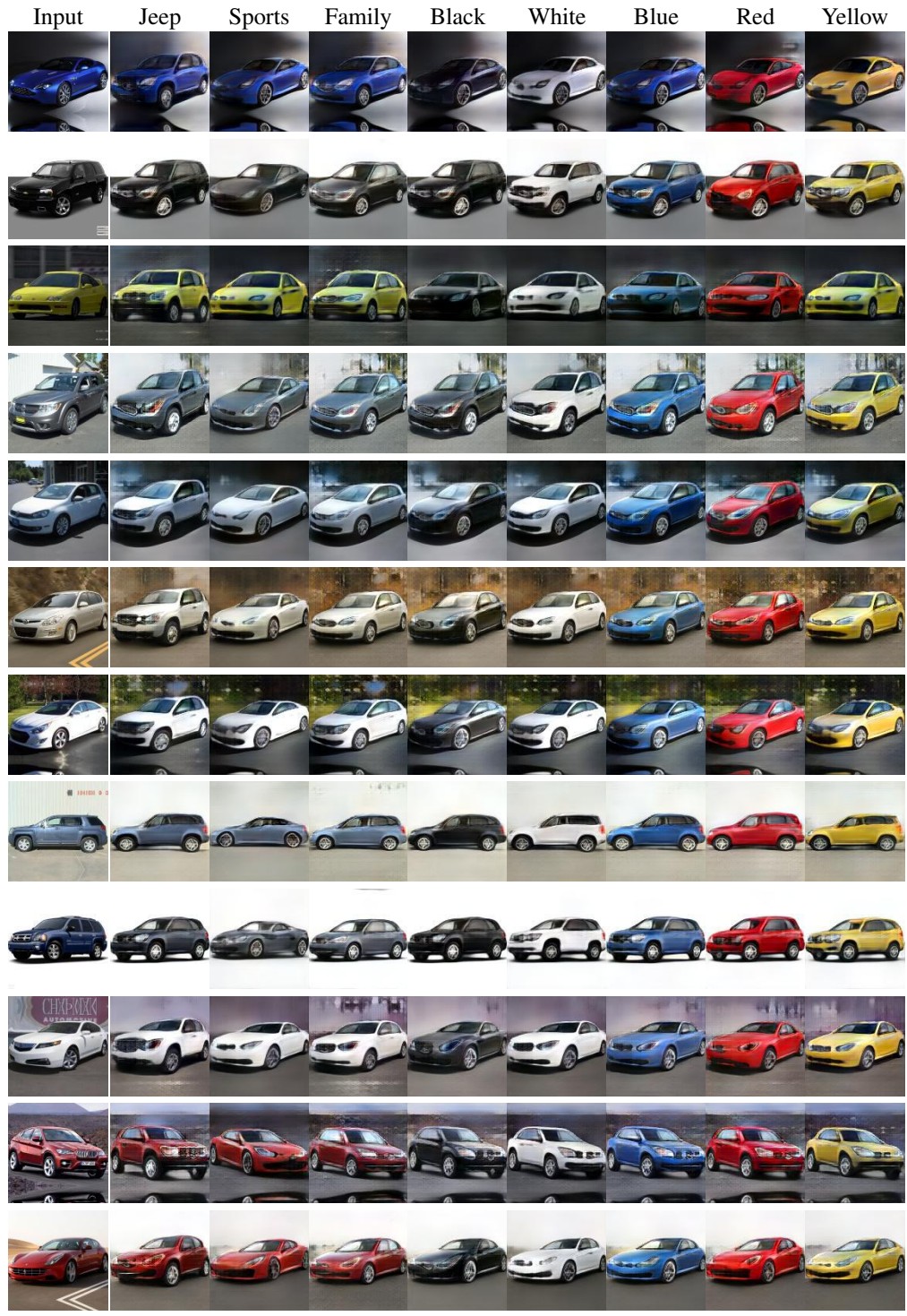

Figure 10: Zero-shot translation results of car types and colors.

Table 10: Values for each attribute used in ZeroDIM for the FFHQ dataset.

| Attribute | Values ($K = 1000$) |
|---|---|
| age | a kid, a teenager, an adult, an old person |
| gender | a male, a female |
| ethnicity | a black person, a white person, an asian person |
| hair | {brunette, blond, red, white, black} hair, bald |
| makeup | makeup, without makeup |
| beard | a person with a {beard, mustache, goatee}, a shaved person |
| glasses | a person with glasses, a person with shades, a person without glasses |

Table 11: Values for each attribute used in ZeroDIM for the AFHQ dataset.

| Attribute | Values ($K = 100$) |
|---|---|
| species | russell terrier, australian shepherd, caucasian shepherd, boerboel, golden retriever, labradoodle, english foxhound, shiba inu, english shepherd, saluki, husky, flat-coated retriever, charles spaniel, chihuahua, dalmatian, cane corso, bengal tiger, sumatran tiger, german shepherd, carolina dog, irish terrier, usa shorthair, lion, snow leopard, lion cat, british shorthair, bull terrier, welsh ke ji, cheetah of asia, himalayan cat, shetland sheepdog, egyptian cat, bombay cat, american bobtail, labrador, american wirehair, chinese li hua, chinese kunming dog, snowshoe cat, maine cat, arctic fox, norwegian forest cat, king shepherd, beagle, ragdoll, brittany hound, tricolor cat, border collie, stafford bull terrier, ground flycatcher, manchester terrier, entrebuche mountain dog, poodle, west highland white terrier, chesapeake bay retriever, hofwald, weimaraner, samoye, hawksbill cat, grey wolf, grey fox, lionesses, singapore cat, african wild dog, yorkshire dog, persian leopard, stacy howler, hygen hound, european shorthair, farenie dog, siberia tiger, jaguar |

Table 12: Values for each attribute used in ZeroDIM for the Cars dataset.

| Attribute | Values ($K = 500$) |
|---|---|
| type | a jeep, a sports car, a family car |
| color | a {black, white, blue, red, yellow} car |

Table 13: betaVAE architecture of the generator $G$ in our synthetic experiments.

| Layer | Kernel Size | Stride | Activation | Output Shape |
|---|---|---|---|---|
| Input | - | - | - | 10 |
| FC | - | - | ReLU | 256 |
| FC | - | - | ReLU | $4 \times 4 \times 64$ |
| ConvTranspose | $4 \times 4$ | $2 \times 2$ | ReLU | $8 \times 8 \times 64$ |
| ConvTranspose | $4 \times 4$ | $2 \times 2$ | ReLU | $16 \times 16 \times 32$ |
| ConvTranspose | $4 \times 4$ | $2 \times 2$ | ReLU | $32 \times 32 \times 32$ |
| ConvTranspose | $4 \times 4$ | $2 \times 2$ | Sigmoid | $64 \times 64 \times$ channels |

Table 14: betaVAE architecture of the classifiers $C^j$ and the residual encoder $E_r$ in our synthetic experiments. $D$ is set to the number of classes $m^j$ of attribute $j$ or the residual dimension $10 - k$.

| Layer | Kernel Size | Stride | Activation | Output Shape |
|---|---|---|---|---|
| Input | - | - | - | $64 \times 64 \times$ channels |
| Conv | $4 \times 4$ | $2 \times 2$ | ReLU | $32 \times 32 \times 32$ |
| Conv | $4 \times 4$ | $2 \times 2$ | ReLU | $16 \times 16 \times 32$ |
| Conv | $2 \times 2$ | $2 \times 2$ | ReLU | $8 \times 8 \times 64$ |
| Conv | $2 \times 2$ | $2 \times 2$ | ReLU | $4 \times 4 \times 64$ |
| FC | - | - | ReLU | 256 |
| FC | - | - | - | $D$ |

Table 15: StyleGAN2-based generator architecture in our experiments on real images. StyleConv and ModulatedConv use the injected latent code which is a concatenation of the representations of the attributes of interest and the residual attributes.

| Layer | Kernel Size | Activation | Resample | Output Shape |
|---|---|---|---|---|
| Constant Input | - | - | - | $4 \times 4 \times 512$ |
| StyledConv | $3 \times 3$ | FusedLeakyReLU | - | $4 \times 4 \times 512$ |
| StyledConv | $3 \times 3$ | FusedLeakyReLU | UpFirDn2d | $8 \times 8 \times 512$ |
| StyledConv | $3 \times 3$ | FusedLeakyReLU | - | $8 \times 8 \times 512$ |
| StyledConv | $3 \times 3$ | FusedLeakyReLU | UpFirDn2d | $16 \times 16 \times 512$ |
| StyledConv | $3 \times 3$ | FusedLeakyReLU | - | $16 \times 16 \times 512$ |
| StyledConv | $3 \times 3$ | FusedLeakyReLU | UpFirDn2d | $32 \times 32 \times 512$ |
| StyledConv | $3 \times 3$ | FusedLeakyReLU | - | $32 \times 32 \times 512$ |
| StyledConv | $3 \times 3$ | FusedLeakyReLU | UpFirDn2d | $64 \times 64 \times 512$ |
| StyledConv | $3 \times 3$ | FusedLeakyReLU | - | $64 \times 64 \times 512$ |
| StyledConv | $3 \times 3$ | FusedLeakyReLU | UpFirDn2d | $128 \times 128 \times 256$ |
| StyledConv | $3 \times 3$ | FusedLeakyReLU | - | $128 \times 128 \times 256$ |
| StyledConv | $3 \times 3$ | FusedLeakyReLU | UpFirDn2d | $256 \times 256 \times 128$ |
| StyledConv | $3 \times 3$ | FusedLeakyReLU | - | $256 \times 256 \times 128$ |
| ModulatedConv | $1 \times 1$ | - | - | $256 \times 256 \times 3$ |

Table 16: StyleGAN2-based discriminator architecture in our experiments on real images.

| Layer | Kernel Size | Activation | Resample | Output Shape |
|---|---|---|---|---|
| Input | - | - | - | $256 \times 256 \times 3$ |
| Conv | $3 \times 3$ | FusedLeakyReLU | - | $256 \times 256 \times 128$ |
| ResBlock | $3 \times 3$ | FusedLeakyReLU | UpFirDn2d | $128 \times 128 \times 256$ |
| ResBlock | $3 \times 3$ | FusedLeakyReLU | UpFirDn2d | $64 \times 64 \times 512$ |
| ResBlock | $3 \times 3$ | FusedLeakyReLU | UpFirDn2d | $32 \times 32 \times 512$ |
| ResBlock | $3 \times 3$ | FusedLeakyReLU | UpFirDn2d | $16 \times 16 \times 512$ |
| ResBlock | $3 \times 3$ | FusedLeakyReLU | UpFirDn2d | $8 \times 8 \times 512$ |
| ResBlock | $3 \times 3$ | FusedLeakyReLU | UpFirDn2d | $4 \times 4 \times 512$ |
| Concat stddev | $3 \times 3$ | FusedLeakyReLU | UpFirDn2d | $4 \times 4 \times 513$ |
| Conv | $3 \times 3$ | FusedLeakyReLU | - | $4 \times 4 \times 512$ |
| Reshape | - | - | - | 8192 |
| FC | - | FusedLeakyReLU | - | 512 |
| FC | - | - | - | 1 |

Table 17: StarGAN-v2-based encoder architecture for the residual attributes in our experiments on real images.

| Layer | Kernel Size | Activation | Resample | Output Shape |
|---|---|---|---|---|
| Input | - | - | - | $256 \times 256 \times 3$ |
| Conv | $3 \times 3$ | - | - | $256 \times 256 \times 64$ |
| ResBlock | $3 \times 3$ | LeakyReLU ($\alpha = 0.2$) | Avg Pool | $128 \times 128 \times 128$ |
| ResBlock | $3 \times 3$ | LeakyReLU ($\alpha = 0.2$) | Avg Pool | $64 \times 64 \times 256$ |
| ResBlock | $3 \times 3$ | LeakyReLU ($\alpha = 0.2$) | Avg Pool | $32 \times 32 \times 256$ |
| ResBlock | $3 \times 3$ | LeakyReLU ($\alpha = 0.2$) | Avg Pool | $16 \times 16 \times 256$ |
| ResBlock | $3 \times 3$ | LeakyReLU ($\alpha = 0.2$) | Avg Pool | $8 \times 8 \times 256$ |
| ResBlock | $3 \times 3$ | LeakyReLU ($\alpha = 0.2$) | Avg Pool | $4 \times 4 \times 256$ |
| Conv | $4 \times 4$ | LeakyReLU ($\alpha = 0.2$) | - | $1 \times 1 \times 256$ |
| Reshape | - | - | - | 256 |
| FC | - | - | - | $D$ |

# F   License of Used Assets

The assets CLIP [35], TediGAN [43] and StyleCLIP [32] use the 'MIT License'.

StyleGAN2 [21] uses the 'Nvidia Source Code License-NC'.