# OpenReview forum: "An Image is Worth More Than a Thousand Words: Towards Disentanglement in The Wild"
_NeurIPS.cc/2021/Conference — NeurIPS 2021 Poster_

### Official Review · Reviewer_X8Vb · 2021-07-13

**Rating:** 6
**Confidence:** 4

**Summary:**

This paper proposes an encoder-decoder based disentanglement method that applies to the semi-supervised and weakly-supervised case (i.e., only a subset of attributes are partially annotated). To adapt the proposed method to real images where the partial labels are unavailable in general, this paper proposes a method of using CLIP to annotate the unlabeled real images, and the resulting image manipulation method is termed ZeroDIM.

Experiments are conducted on synthetic datasets (Shapes3D, Cars3D, dSprites and SmallNORB) to show better performance on disentanglement learning, and experiments are conducted on real datasets (FFHQ, AFHQ and Cars) to show better image manipulation results.


**Ethical Concerns:**

Because the paper studies deep learning methods on image manipulation, there are some concerns regarding its malicious use and potential bias in the real life. The authors have well stated them and attributed the bias to the use of CLIP.

**Limitations And Societal Impact:**

The authors have adequately addressed the limitations and potential negative societal impact of their work

**Main Review:**

Strengths:
+ This paper is well-written, the proposed method is easy to understand and the experiment settings are clearly stated.
+ The latent optimization for better disentangling the residual attributes from the considered attributes of interest is interesting.
+ The idea of using CLIP to partially label the real images in a zero-shot way is very interesting.
+ The proposed method outperforms other disentangled methods as measured by DCI, SAP and MIG.


Weaknesses:

- The quantitative evaluation on real images is missing. In the paper, only visual examples of different image editing methods are provided, which I think could be biased and subjective given that the proposed method is not visually much better than the baselines (TediGAN and StyleCLIP). I would recommend using some metrics to measure the image editing quality (such as disentanglement, FID and controllability) or doing some human study evaluations.
-  For the proposed disentanglement model, I do not see much novelty on how to deal with partially labeled attributes (Eq. 3-5), since as far as I know, previous works have already used this similar idea, such as [1].
- I am just curious about why Locatello et al. [29] has such low MIG scores on these benchmarks (say, 0.01 on dSprites). Is this really because of permuted-labels configuration or the hyperparameters in the baseline have not been well selected?
- It looks like the highest resolution of real images in this paper is only 256x256, which seems to limit the generation quality of the proposed method. I wonder if the method can be scaled up to a larger resolution, such as 1024x1024 in the StyleGANs. Is there any constraint in doing so?


[1] Kingma et al., Semi-Supervised Learning with Deep Generative Models, NeurIPS 2014.



=================

Post-rebuttal update:

I thank the authors for taking the time to respond to my reviews. But my major concerns still remain: 1) The qauntitative results of real images actually show the worse disentanglement performance measured by AD (though I agree that it only refects one aspect of disentanglement quality). 2) Although [1] only studies the single-attribute case, Eq. (3) and (4) look exactly the same with the second term of Eq. (7) and the second term of Eq. (9) in [1]. Thus, I still think the novelty of dealing with partially labeled attributes is limited. 3) The visual quality on real images (up to res 256x256) is not high, so I'm a little doubtful that the proposed method can be scaled to more complex high-res image. Therefore, I keep the initial rating unchanged and tend to reject.


=================

Post-rebuttal update (2nd round):

I thank the reviewer for the further clarifications, in particular about the difference between this work and the reference [1]. I agree that disentanglement evaluation on real images is more challenging but I think solely reporting AD results (i.e., measuring how the modification affects other attributes) is problematic. I suggest the authors also consider other metrics to measure the "editing strength" (i.e., how large the modification changes the considered attribute). One implementation could be that we first train image attribute classifiers on CelebA and use them as oracle models to measure the predicted score differences of generated images before and after editing. Overall, I'm convinced by the authors regarding the novelty and happy to raise my initial score by 1.


**Time Spent Reviewing:**

4

---

> ### Author Response · Authors · 2021-08-10
> **Response to Reviewer X8Vb**
>
> We thank the reviewer for the dedicated review and for finding our method “very interesting”.
> ___
>
> **"The quantitative evaluation on real images is missing"**: The quantitative evaluation of disentanglement in real images is challenging as no ground truth annotations are available for all the attributes and the attributes are not completely independent. For evaluation purposes, the paper includes quantitative metrics on synthetic benchmark and many qualitative comparisons on real images. However, we made an effort to address your concern and consider quantitative metrics for evaluation of our method on real images of human faces. We assess the performance by Attribute-Dependency (AD) (proposed in [1]): we measure the degree to which manipulation of a certain attribute induces changes in other attributes, as measured by classifiers for those attributes. We rely on 40 pretrained classifiers for attributes in CelebA, in order to cope with real images, where the exact factors of variation are not observed. Intuitively, disentangled manipulations should induce smaller changes in other attributes (lower AD is better). The following table shows the AD scores of all methods while manipulating different attributes of interest.
>
> [1] Wu et al. StyleSpace Analysis: Disentangled Controls for StyleGAN Image Generation. In CVPR, 2021.
>
> | Manipulated attribute: |  Age | Beard | Ethnicity | Gender | Glasses | Hair Color |
> |:---------------------:|:----:|:-----:|:---------:|:------:|---------|------------|
> | TediGAN               | 0.39 | 0.38  | 0.41      | 0.40   | 0.31    | 0.37       |
> | StyleCLIP             | 0.45 | 0.42  | 0.40      | 0.78   | 0.35    | 0.44       |
> | LORD [14]             | 0.41 | 0.65  | 0.38      | 0.36   | 0.46    | 0.38       |
> | Ours                  | 0.40 | 0.36  | 0.40      | 0.44   | 0.49    | 0.37       |
>
> We stress that quantitative measurements of this sort are not perfect and can sometimes be misleading. However, let us briefly review the main trends reflected by these metrics (the same trends are clearly visualized in Fig. 5-7 in the appendix): (i) StyleCLIP tends to over manipulate the desired attribute and causes changes to other attributes of the input image, resulting in inferior disentanglement and leading to higher AD scores. This can be clearly seen when changing gender or hair color. On the other side, some manipulations seem negligible (e.g. invisible glasses), leading to low AD scores. (ii) LORD struggles to disentangle attributes which are not perfectly uncorrelated e.g. manipulating gender does not affect the input image at all (leading to low AD scores) while adding beard to females leads to gender swapping (higher AD scores). Note that TediGAN mostly introduces artifacts without manipulating the desired attribute, and therefore maintains low AD scores.
>
> **"For the proposed disentanglement model, I do not see much novelty on how to deal with partially labeled attributes (Eq. 3-5), since as far as I know, previous works have already used this similar idea, such as [1]."**: [1] is a seminal paper in semi-supervised generative modeling. While this paper indeed tackles a similar semi-supervised setting, there is a major limitation in applying their proposed approach to the tasks considered in our paper. Recall that we aim to disentangle several different attributes of interest, each of which can take one out of many different values. A limitation of the models presented in [1] is that they scale linearly with the number of possible values of each attribute of interest, since the generative likelihood is re-evaluated for each possible value in each sample in the mini-batch during training (Eq. 7 in [1]). This exhaustive inference is clearly an expensive operation which inevitably limits the applicability of the approach (see the original discussion section in [1]). Extending these models to multi-attribute disentanglement is not trivial, as the number of attribute assignments that is needed to be evaluated during training grows exponentially. For example, even for a synthetic benchmark as dSprites, enumerating all the assignments of three attributes of interest (scale, x, y) results in 6x32x32=6144 passes through the generative model for each sample in the training batch to derive the gradients! Our latent optimization based approach for learning the residual attributes (differently from conditioning on the attributes of interest as in [1]) greatly reduces the computational cost while obtaining disentangled representations.  We will refer to [1] in our related work section, although it can not serve as a baseline in our experimental study.
>
> **"I am just curious about why Locatello et al. [29] has such low MIG scores on these benchmarks (say, 0.01 on dSprites). Is this really because of permuted-labels configuration or the hyperparameters in the baseline have not been well selected?"** Locatello et al. struggle with the considered setting for two reasons: (i) The presence of residual factors of variation which are never observed. (ii) The permuted-labels configuration (as you hypothesized), in which the attribute values are not assumed to exhibit a semantic order. While the order of the values can be exploited as an inductive bias to better disentanglement, many attributes in the real world (e.g. human gender or animal species) are not ordered in a meaningful way, which is the main objective of our work. We have experimented with several hyperparameter sets (suggested in the appendix of [29]) and failed to obtain significantly better results.
>
> **"It looks like the highest resolution of real images in this paper is only 256x256, which seems to limit the generation quality of the proposed method. I wonder if the method can be scaled up to a larger resolution, such as 1024x1024 in the StyleGANs."**: The highest resolution we consider is indeed 256x256. We have not experimented yet with images at 1024x1024 due to limited computational resources. However, we do not expect any fundamental barrier in scaling up our method to higher resolution.

---

> ### Author Response · Authors · 2021-08-26
> **Additional Response to Reviewer X8Vb**
>
> Thank you for taking the time for responding to our comments. We hope that our additional clarifications will resolve your remaining concerns.
> ___
>
> **AD results:** We believe the reviewer may be referring to the better (i.e. lower) AD scores of TediGAN. As clearly seen in Fig. 5-7 in the appendix, TediGAN makes minor changes to the desired and undesired attributes.  Quantitatively, this leads to misleading, low AD scores but does not achieve the main objective of correct image manipulation. This highlights a well known issue (see Reviewer bP97) of quantitative evaluation metrics for disentanglement in real images. While it may seem that StyleCLIP's performance is not much worse than ours in disentangling *local* concepts (due to the benefits of StyleGAN architectural biases), the evaluation shows that the disentanglement of *global* concepts (e.g. Gender) is much better with our method.
>
> **Technical novelty in relation to [1]:** While the method presented in [1] addresses a similar setting, there are three important novelties in our paper:
>
> - The method in [1] can only address the setting of disentangling a single low-cardinality attribute while we can address the setting of disentanglement of multiple attributes with high-cardinality (this limitation of [1] is admitted by the original authors of [1] in their discussion section).
>
> - Following this different setting, the approach taken by [1], i.e. conditioning the residuals on the attributes of interest and re-evaluating the model for each possible attribute value are no longer feasible. Hence, we have major technical differences; (a) We do not directly compute the expectation of the reconstruction error of the generator over all the possible values of the attributes of interest, and our residual encoder is also not conditioned on the attributes of interest. Instead, we merely compute the expectation over the latent codes of each attribute of interest (i.e. multiplying the output probability vector of classifier $j$ by the embeddings of attribute of interest $j$ within the generator, as detailed in appendix A.1). We compute the residual code with latent optimization instead of conditioning on the attributes of interest (the reviewer noted that this is an interesting approach). We then compute the reconstruction loss only once. This yields exponential efficiency gains in the case of multiple attribute of interest. (b) Our regularization term minimizes the entropy of $p(y|x)$ rather than maximizing it. In other words, our loss term encourages the classifier to be as certain of its label as possible, whereas Eq.7 in [1] encourages the probabilities to be as spread out as possible (which is a very different prior that comes from the ELBO criterion). Please see response to Reviewer VXQA for an ablation of the benefit of our loss term. Also, [1] assumes that all the attribute values are balanced, which is clearly not the case for all attributes e.g. hair color in human faces. In view of the above, we insist that despite some similarities, our loss terms are significantly different from those in [1].
>
> - The benchmarks explored in [1] were purely synthetic. We further scale our approach to real images, as well as utilizing off-the-shelf image-language embedding models as CLIP for partial annotations, without manual effort.
>
> For these reasons we believe the contributions of our work may be of great interest to the disentanglement community. We will make this clearer in the final version of the paper.
>
> **Image resolution:** Our experiments are conducted at a relatively high-resolution (up to 256x256) for the disentanglement community. The vast majority of methods in the disentanglement literature only consider low-res (64x64) and synthetic images, e.g. the comprehensive study presented in Locatello et al. [29]. Very recent high-resolution disentanglement papers e.g. OverLORD [15] are evaluated at up to 256x256 resolution, similarly to our work. In general, the task of disentanglement is sufficiently challenging and mostly orthogonal to synthesizing images at higher resolutions (e.g. 1024x1024). We believe that many of the ideas in synthesizing high-resolution images can be incorporated in our method (we have already borrowed the architecture of StyleGAN2), but this is not the main objective of our work, and is beyond our computational resources.

---

> ### Author Response · Authors · 2021-08-27
> **Thank you for the updated score!**
>
> Thank you for acknowledging our clarifications and for the positive reconsideration of our paper. We believe that measuring the "editing strength" is a good idea, we will incorporate this additional metric in the final version.

---

### Official Review · Reviewer_VXQA · 2021-07-15

**Rating:** 6
**Confidence:** 3

**Summary:**

The paper investigates the task of disentanglement under partially labelled data. The setting is well-motivated as new general-purpose models like CLIP emerge, which can be used to provide partial labelling.  The proposed formulation is intuitive, utilizes supervised training on the labelled attributes, an entropy regularizer on the unlabelled ones, and a reconstruction loss. Experiment results on standard datasets (e.g., Shapes3D, dSprites) confirm the effectiveness of the method. The authors extend the method to a zero-shot setting, where partial labels are obtained via CLIP latent similarity matching. Qualitative results on real-image datasets (e.g., FFHQ, AFHQ, cars) show the advantage over various baselines, including StyleGAN-based methods.

**Limitations And Societal Impact:**

The authors have addressed the limitations and potential societal impact in Sections 4.3 and 5.

**Main Review:**

The paper is clearly written. The partially labelled setting is well-motivated from the perspective of utilizing models like CLIP. The proposed method is intuitive, and the results seem promising. I have one questions regarding to the method:

1. What's the motivation & effect of adding a Gaussian noise for the residual code?

For experiments, while the authors provide results on many datasets, I find the paper lacking a couple more baselines and evaluations.

1. Since LORD requires a supervision, another straightforward baseline is to train LORD based on pseudo-labels given by the classifier described in Section 3.4.
2. Could the authors provide some more quantitative evaluation on the manipulation results? This can be obtained e.g. via attribute classifiers and/or human evaluation.
3. The LORD manipulations shown in the appendix are of quite high quality. From my visual perception, there are cases where LORD performs well, or the proposed method performs well. While it seems that the proposed method may doing consistently a better job, I feel that LORD results should definitely be included in main text, along with the quantitative evaluations (mentioned above) to compare it with the proposed method.
4. Could the authors provide ablation & hyperparameter studies on the proposed method (e.g., the effect of Gaussian noise regularization on the residual code)? In the current paper, it is unclear the significance of some of these components, and the sensitivity to hyperparameters. The appendix claims that its Table 4 is an ablation study, but it really is just a complete comparison to pseudo-labels baseline.

If the authors can address my concerns, I am willing to raise my score.


---
Update (8/14): the authors' response has addressed my comments and I have raised the evaluation accordingly.

**Time Spent Reviewing:**

2

---

> ### Author Response · Authors · 2021-08-10
> **Response to Reviewer VXQA**
>
> We thank the reviewer for the dedicated review and for recognizing our “intuitive” method and “promising” results. We hope that the following clarifications address the concerns and would indeed form a basis for raising the score.
> ___
>
> **"What's the motivation and effect of adding a Gaussian noise for the residual code?"**: As the residual code is idiosyncratic to each image, the network may naively store the entire information on each image in the residual code and ignore the attributes codes altogether. In this case the residual code will not be disentangled from the attributes. As we would like the residual code to contain only the information on each image which cannot be coded in the attributes of interest, we are aiming to limit the information content which can be used to describe an individual image. This regularization form was properly justified in [14]. We will briefly explain it here (and in the final version of the paper for completeness). As the residual attributes are completely unsupervised, we regularize the respective latent variables, similarly to variational auto-encoders (VAEs) i.e. matching their distribution to a prior Normal distribution (with zero mean, and unit covariance). The term simply translates to:
> \begin{equation}
> D_{KL}(\mathcal{N}(\mu, \sigma^2 I) || \mathcal{N}(0, I)) = -\frac{1}{2} \sum_d (\ln{\sigma_d^2}-\sigma_d^2-\mu_d^2 + 1)
> \end{equation}
> Where $\mu$ and $\sigma$ vectors express the mean and standard deviation of the posterior latent distribution of the residual attributes (given an input image $x$). According to a key finding presented in [14], we do not learn the variance of the posterior distribution but rather set it fixed to  $\sigma_d^2=1$. The regularization term therefore translates to minimizing $||\mu||^2$, as stated in Eq. (6). Similarly to VAE, this sets a bottleneck on the amount of information in the latent code.
>
> **"Since LORD requires supervision, another straightforward baseline is to train LORD based on pseudo-labels given by the classifier described in Section 3.4."**: As per you suggestion, we annotate all the images in the dataset with pseudo-labels given by the classifier in Section 3.4, and train LORD [14] as a fully specified model (except for the residual attributes, as usual). The final disentanglement performance is sub-optimal to our method, as summarized below (using 1000 [or 100] labels per attribute of interest).
>
> |           |             |      D      |      C      |      I      |     SAP     | MIG         |
> |:---------:|:-----------:|:-----------:|:-----------:|:-----------:|:-----------:|-------------|
> | Shapes3D  | LORD-pseudo | 1.00 [0.87] | 1.00 [0.87] | 1.00 [0.87] | 0.30 [0.20] | 1.00 [0.79] |
> | Shapes3D  | Ours        | 1.00 [1.00] | 1.00 [1.00] | 1.00 [1.00] | 0.30 [0.30] | 1.00 [0.96] |
> | Cars3D    | LORD-pseudo | 0.78 [0.38] | 0.78 [0.39] | 0.78 [0.43] | 0.30 [0.12] | 0.61 [0.28] |
> | Cars3D    | Ours        | 0.80 [0.40] | 0.80 [0.41] | 0.78 [0.56] | 0.33 [0.15] | 0.61 [0.35] |
> | dSprites  | LORD-pseudo | 0.86 [0.57] | 0.86 [0.57] | 0.46 [0.36] | 0.12 [0.09] | 0.48 [0.34] |
> | dSprites  | Ours        | 0.91 [0.75] | 0.91 [0.75] | 0.69 [0.68] | 0.14 [0.13] | 0.57 [0.48] |
> | SmallNorb | LORD-pseudo | 0.61 [0.29] | 0.63 [0.34] | 0.48 [0.40] | 0.20 [0.16] | 0.37 [0.27] |
> | SmallNorb | Ours        | 0.63 [0.27] | 0.65 [0.39] | 0.53 [0.45] | 0.20 [0.14] | 0.40 [0.27] |
>
> **"Could the authors provide some more quantitative evaluation on the manipulation results? This can be obtained e.g. via attribute classifiers and/or human evaluation"**: The quantitative evaluation of disentanglement in real images is challenging as no ground truth annotations are available for all the attributes and the attributes are not completely independent. For evaluation purposes, the paper includes quantitative metrics on synthetic benchmark and many qualitative comparisons on real images. However, we made an effort to address your concern and consider quantitative metrics for evaluation of our method on real images of human faces. We assess the performance by Attribute-Dependency (AD) (proposed in [1]): we measure the degree to which manipulation of a certain attribute induces changes in other attributes, as measured by classifiers for those attributes. We rely on 40 pretrained classifiers for attributes in CelebA, in order to cope with real images, where the exact factors of variation are not observed. Intuitively, disentangled manipulations should induce smaller changes in other attributes (lower AD is better). The following table shows the AD scores of all methods while manipulating different attributes of interest.
>
> [1] Wu et al. StyleSpace Analysis: Disentangled Controls for StyleGAN Image Generation. In CVPR, 2021.
>
> | Manipulated attribute: |  Age | Beard | Ethnicity | Gender | Glasses | Hair Color |
> |:---------------------:|:----:|:-----:|:---------:|:------:|---------|------------|
> | TediGAN               | 0.39 | 0.38  | 0.41      | 0.40   | 0.31    | 0.37       |
> | StyleCLIP             | 0.45 | 0.42  | 0.40      | 0.78   | 0.35    | 0.44       |
> | LORD [14]             | 0.41 | 0.65  | 0.38      | 0.36   | 0.46    | 0.38       |
> | Ours                  | 0.40 | 0.36  | 0.40      | 0.44   | 0.49    | 0.37       |
>
> We stress that quantitative measurements of this sort are not perfect and can sometimes be misleading. However, let us briefly review the main trends reflected by these metrics (the same trends are clearly visualized in Fig. 5-7 in the appendix): (i) StyleCLIP tends to over manipulate the desired attribute and causes changes to other attributes of the input image, resulting in inferior disentanglement and leading to higher AD scores. This can be clearly seen when changing gender or hair color. On the other side, some manipulations seem negligible (e.g. invisible glasses), leading to low AD scores. (ii) LORD struggles to disentangle attributes which are not perfectly uncorrelated e.g. manipulating gender does not affect the input image at all (leading to low AD scores) while adding beard to females leads to gender swapping (higher AD scores). Note that TediGAN mostly introduces artifacts without manipulating the desired attribute, and therefore maintains low AD scores.
>
> **"I feel that LORD results should definitely be included in main text."**: As per your suggestion, we will move the qualitative comparison with LORD from the appendix to the main text, along with the quantitative evaluation provided in this rebuttal.
>
> **"Could the authors provide ablation and hyperparameter studies on the proposed method e.g., the effect of Gaussian noise regularization on the residual code"**: We provide an ablation study of the different terms in Eq. (8), as per your request.
>
> - ***Ablation for $\mathcal{L}_{ent}$ on Shapes3d, using 1000 [or 100] labels per attribute of interest***
>
> |                              |      D      |      C      |      I      |     SAP     |     MIG     |
> |------------------------------|:-----------:|:-----------:|:-----------:|:-----------:|:-----------:|
> | Ours w/o $\mathcal{L}_{ent}$ | 0.99 [0.99] | 0.98 [0.98] | 0.98 [0.98] | 0.28 [0.26] | 0.94 [0.91] |
> | Ours                         | 1.0 [1.0]   | 1.0 [1.0]   | 1.0 [1.0]   | 0.30 [0.30] | 1.0 [0.96]  |
>
> - ***Ablation for $\mathcal{L}_{ent}$ on Cars3d, using 1000 [or 100] labels per attribute of interest***
>
> |                              |       D      |      C      |      I      |     SAP     | MIG         |
> |----------------------------|:------------:|:-----------:|:-----------:|:-----------:|-------------|
> | Ours w/o $\mathcal{L}_{ent}$ | 0.74 [0.39]  | 0.74 [0.40] | 0.71 [0.43] | 0.22 [0.11] | 0.57 [0.34] |
> | Ours                         | 0.80 [0.40]  | 0.80 [0.41] | 0.78 [0.56] | 0.33 [0.15] | 0.61 [0.35] |
>
> - ***Ablation for $\mathcal{L}_{res}$ on Shapes3d, using 1000 labels per attribute of interest*** (accuracy of classifying the attributes of interest from the residual representations - lower is better)
>
> |                               | floor_color | wall_color | object_color |
> |-----------------------------|:----------:|:---------:|:-----------:|
> | Ours w/o $\mathcal{L}_{res}$  | 0.23       | 0.28      | 0.18        |
> | Ours                          | 0.11       | 0.12      | 0.14        |
> | Random Chance (optimal)       | 0.1        | 0.1       | 0.1         |
>
> The entire ablation study would be added to the final version.

---

> > ### Comment · Reviewer_VXQA · 2021-08-14
> > **Re: Response to Reviewer VXQA**
> >
> > Thanks for the response and detailed experimental results. I have updated my score accordingly.

---

> > > ### Author Response · Authors · 2021-08-20
> > > **Thank you for the updated score!**
> > >
> > > Thank you for acknowledging our response and for the positive reconsideration of our paper.

---

### Official Review · Reviewer_bP97 · 2021-07-16

**Rating:** 8
**Confidence:** 4

**Summary:**

Proposed a method for disentangling a set of partially labeled factors while also capturing all unlabelled factors. As an application, CLIP is used to annotate a subset of attributes in natural images. Disentanglement is evaluated using the proxy task of image manipulation.

**Limitations And Societal Impact:**

The authors adequately discussed limitations and societal impact.

**Main Review:**

# Originality
The novel contribution is a method for disentanglement in the setting where only a subset of images is labeled partially (i.e. different labels are available for different images). This differs from prior work such as Locatello et al. [29] which assume that all labels are available in each labelled images. The authors demonstrate the practicality of such a setting with their zero-shot disentangled image manipulation experiments using the recent CLIP model.

# Quality
While many details as well as results have been pushed to the appendix, the work appears fairly complete. A few points of note:

(1) Locatello et al. [28] demonstrated the significance of random seed in previous disentanglement works, however it does not appear as if results are summarized across multiple random seeds. It would be good to perform significance tests, especially for the results in Table 2 where differences are relatively small.

(2) Factors which are held-out as residuals are not evaulated with respect to. For example, on Shapes3D, scale, shape, and azimuth were denoted as residuals, thus the authors did not evaluate whether said factors could be disentangled by their model. Notably on dSprites, orientation and shape were held out, the factors on dSprites which previous work [23] has notably shown are significantly harder for existing disentanglement methods than the labeled attributes (scale, x, y). I understand that the setting in Table 1 is interesting to check information leakage into the residuals, but providing results for all attributes (and no residuals) would make the disentanglement results much more comparable to existing work.

(3) The zero-shot disentangled image manipulation results are purely qualitative. While the results do not appear to be obviously cherry-picked, there is a bit of ambiguity whether these figures demonstrate that the proposed model is *significantly* better than the given baselines. I understand that there basically is no quantitative metric to measure disentanglement, but more comparisons e.g. with respect to latent traversals (e.g. increasing the "asian" latent from low to high) on randomly chosen images would be welcome.

# Clarity
The manuscript is well-written and organized. I would have appreciated a more detailed comparison to related work. It is mentioned in passing that many implementation details were borrowed from existing methods, namely LORD [14] and OverLORD [15], so an extended discussion, possibly in the appendix, where the detailed differences between the proposed system and said existing systems would be appreciated.

# Significance
The assumed setting of partially labeled images is of practical relevance. Demonstrating the applicability to natural images is definitely welcome and while the evaluation is purely qualitative, showing how CLIP can be used for partial labelling and disentanglement is an interesting idea.

# Rebuttal
I thank the authors for their very extensive and detailed response. I raise my score by one point.

**Time Spent Reviewing:**

3

---

> ### Author Response · Authors · 2021-08-10
> **Response to Reviewer bP97**
>
> We thank the reviewer for the dedicated and positive review.
> ___
>
> **"Locatello et al. [28] demonstrated the significance of random seed in previous disentanglement works, however it does not appear as if results are summarized across multiple random seeds."**: Reproducing an extensive study as presented in Locatello et al. requires approximately 8.57 GPU years (23,040 models in total). However, as per your request, we made an effort and run our smaller set of synthetic experiments with $5$ different random seeds (for the supervised labels and the initialization of the model). The results are pretty consistent i.e. all the metrics exhibited a standard deviation of 0.001 to 0.01 in the different trials. We will update the relevant tables in the final version to be inline with other disentanglement papers.
>
> **"Providing results for all attributes (and no residuals) would make the disentanglement results much more comparable to existing work."**: As per your suggestion, we conducted experiments on the synthetic datasets with all the factors of variation treated as attributes of interest, holding out no residual factors at all, which is actually the setting studied in Locatello el al [29]. While this is not the task we aim to solve, our method performs better than Locatello et al [29] using the same beta-vae based architecture. The results using 1000 labels per attribute are summarized below (we show mean [std] over 5 random seeds).
>
>
> |           |           | DCI Disentanglement |      SAP     |     MIG     |
> |:---------:|:---------:|:-------------------:|:------------:|:-----------:|
> | Shapes3D  | Locatello [29] | 0.99 [0.001]        | 0.23 [0.01]  | 0.75 [0.05] |
> | Shapes3D  | Ours      | 1.00 [0.001]        | 0.37 [0.001] | 0.99 [0.01] |
> | Cars3D    | Locatello [29] | 0.58 [0.05]         | 0.14 [0.01]  | 0.25 [0.01] |
> | Cars3D    | Ours      | 0.59 [0.06]         | 0.19 [0.01]  | 0.49 [0.01] |
> | dSprites  | Locatello [29] | 0.46 [0.03]         | 0.07 [0.001] | 0.33 [0.01] |
> | dSprites  | Ours      | 0.62 [0.01]         | 0.09 [0.01]  | 0.39 [0.01] |
> | SmallNorb | Locatello [29] | 0.43 [0.02]         | 0.13 [0.01]  | 0.24 [0.01] |
> | SmallNorb | Ours      | 0.68 [0.01]         | 0.31 [0.02]  | 0.52 [0.01] |
>
> **"The zero-shot disentangled image manipulation results are purely qualitative."**: The quantitative evaluation of disentanglement in real images is indeed challenging as no ground truth annotations are available for all the attributes and the attributes are not completely independent. For evaluation purposes, the paper includes quantitative metrics on synthetic benchmark and many qualitative comparisons on real images. However, we made an effort to address your concern and consider quantitative metrics for evaluation of our method on real images of human faces. We assess the performance by Attribute-Dependency (AD) (proposed in [1]): we measure the degree to which manipulation of a certain attribute induces changes in other attributes, as measured by classifiers for those attributes. We rely on 40 pretrained classifiers for attributes in CelebA, in order to cope with real images, where the exact factors of variation are not observed. Intuitively, disentangled manipulations should induce smaller changes in other attributes (lower AD is better). The following table shows the AD scores of all methods while manipulating different attributes of interest.
>
> [1] Wu et al. StyleSpace Analysis: Disentangled Controls for StyleGAN Image Generation. In CVPR, 2021.
>
> | Manipulated attribute: |  Age | Beard | Ethnicity | Gender | Glasses | Hair Color |
> |:---------------------:|:----:|:-----:|:---------:|:------:|---------|------------|
> | TediGAN               | 0.39 | 0.38  | 0.41      | 0.40   | 0.31    | 0.37       |
> | StyleCLIP             | 0.45 | 0.42  | 0.40      | 0.78   | 0.35    | 0.44       |
> | LORD [14]             | 0.41 | 0.65  | 0.38      | 0.36   | 0.46    | 0.38       |
> | Ours                  | 0.40 | 0.36  | 0.40      | 0.44   | 0.49    | 0.37       |
>
> We stress that quantitative measurements of this sort are not perfect and can sometimes be misleading. However, let us briefly review the main trends reflected by these metrics (the same trends are clearly visualized in Fig. 5-7 in the appendix): (i) StyleCLIP tends to over manipulate the desired attribute and causes changes to other attributes of the input image, resulting in inferior disentanglement and leading to higher AD scores. This can be clearly seen when changing gender or hair color. On the other side, some manipulations seem negligible (e.g. invisible glasses), leading to low AD scores. (ii) LORD struggles to disentangle attributes which are not perfectly uncorrelated e.g. manipulating gender does not affect the input image at all (leading to low AD scores) while adding beard to females leads to gender swapping (higher AD scores). Note that TediGAN mostly introduces artifacts without manipulating the desired attribute, and therefore maintains low AD scores.
>
> **"Detailed comparison to related work (e.g. LORD)"**: Our method can be seen as an extension of LORD [14] for cases where the attributes of interest are observed only in a very few samples. From a technical perspective, there are two fundamental differences between our method and LORD [14]: (i) LORD is a fully latent-based model i.e. no classifiers are trained with the generator in the first stage. The latent codes of the attributes of interest are optimized directly (and shared between all instances with the same label). Here we provide a hybrid latent-amortized approach where codes are learned in a latent fashion, similarly to LORD, but they are weighted using the probabilities emitted by an amortized classifier.  (ii) We introduce an additional term $\mathcal{L}_{ent}$ which enables our method to perform well when very limited supervision exists for the attributes of interest. We show that our method strongly outperforms [14] in the few-labels setting, achieving better quantitative scores and fundamentally more disentangled qualitative manipulations on real images. Moreover, the robustness of our method to partial-labeling facilitates the use of general-purpose classification models such as CLIP for zero-shot image manipulation.

---

> > ### Comment · Reviewer_bP97 · 2021-08-13
> > **Thanks!**
> >
> > Thanks for the detailed and extensive response! I am raising my score by one point.

---

> > > ### Author Response · Authors · 2021-08-20
> > > **Thank you for the updated score!**
> > >
> > > Thank you for acknowledging our response and for the positive assessment of our work.

---

### Official Review · Reviewer_Z3et · 2021-07-20

**Rating:** 6
**Confidence:** 4

**Summary:**

The paper presents a new approach for disentangling attributes from images in a semi-supervised setting, where attributes are only available for a subset of images, and only a subset of attributes are observed. The basic idea is to use as a latent space output from a classifier trained on images, along with a residual latent space and use both of them to reconstruct an image. Inference is done either via direct optimization with respect to the learnt generator or by fitting a second-stage inference network using the learnt generator. The empirical studies appear to show that the proposed method improves over previous methods [29] and [14].


**Limitations And Societal Impact:**

I find the discussion about the limitations and broader impacts to be adequate.

**Main Review:**

The high-level question asked by the work is very exciting and relevant to the research in disentangling, the execution in terms of formulation experiments, explaining modeling choices, technical correctness and clarity, and performing proper ablations on the models could be improved substantially. The major concerns I have with the work are all marked with a (*).

### Strengths

- The problem of underspecification when it comes to real-image datasets is well motivated with respect to disentangling literature. A clean solution to this problem would indeed have real impact.
- The paper has nice qualitative results on real images

### Weaknesses

1. **Framing and Novelty of the studied problem**
    - While the framing of the problem and motivation is neat, the execution of the high-level idea is not as interesting a scientific question to study, in my opinion. What I mean by this is that it would have been interesting to study if one could disentangle some attributes given supervision of some other attributes to uncover the mechanisms of the world. For example, consider being given some of the basis vectors for a representation, then one could imagine figuring out what other basis vectors /could/ be by reasoning that all the given basis vectors are for example orthogonal. This would indeed have been a nice inter-mingling of the unsupervised disentangling and supervised disentangling directions that have been pursued in the literature.
    - However, the specific problem tackled by the current paper is far less impactful given the high-level, general direction of the paper, in the sense that the manner in which the paper approaches this problem, that is, disentangling what is supervised and packing the rest into a different vector is not very different from previous work on style and content disentangling [14, 4] etc. (with the only minor difference being that the content is also disentangled). (*)

2. **Methods:**
    - The general idea of the paper is very similar to that of [14], with the only difference that class annotations in [14] are replaced with a set of attribute annotations, and the residual vector in the current work is the content vector of [14]. With this mapping of terminologies, it is unclear what the novelty of the proposed approach is, and appears rather incremental. (*)
    - Beyond this, there are a number of half-baked technical claims which are either incorrect or unclear (the distinction is not obvious to me in all the cases). For example, the paper claims that one can restrict the information in the labels by maximizing the entropy of the label distribution from the attribute classifier given an image. I imagine one wants to stop the information from the image from leaking into the classifier? If so, denoting by Y the attribute labels and X the image, the (Shannon) mutual information between X and Y is H(X) - H(Y| X). Thus, if one wanted the labels Y to have low information about the image, one would actually maximize the conditional entropy not minimize it! (*)
    - Another claim is that the information in a gaussian random variable can be minimized by regularizing the mean to have a low L2-norm. Even if one just looked at the entropy of a gaussian (and not the conditional entropy) and assumed that someone minimizing it was the right thing to do (which the previous bullet point indicates might not be), the differential entropy of a gaussian [A] actually does not even depend on the mean, and only depends on the variance. Thus, in some sense the norm of the means has nothing to do with the information in the gaussian distribution.  (*)

3. **Minor suggestions on clarity:**
    - L132 talks about the output of inference without explaining how inference is done. Would be nice to show / indicate the inference procedure in Fig. 1 to fix this.
    - Given how closely related the approach seems to be to [14] it would really help to have a concrete discussion of how the proposed approach and task are different from [14] in the related work section. (*)

4. **Empirical Results:**
    - It is really hard to understand why the proposed method does better than [14] in Table. 1, since it is essentially identical. Are attribute classifiers not trained for [14]? If so, what is the step that contributes to the gains of the proposed method? Is it the regularization of the softmax output that is done in this method but not in [14]? It would be great to see an ablation of this term to establish this.  (*)
    - In general, it would be helpful to see an ablation study of the contribution of each of the terms in Eqn. (8) to the performance. Without this, given the state of the art results it is really difficult to trust and understand how the performance came about, given that the method itself is conceptually very similar to prior work [14]. (*)
    - Is the evaluation only done on the specified attribute subset or for all the factors of variation in the dataset? Please clarify. If the former, it would be great to include results for the latter as well (perhaps in a major revision)
    - Table 2. It is not clear / not described what the pseudo-labels method here is.
    - In general, it is very unclear how Sec. 4 results fit into the rest of the paper. While it is great that real image results are presented, it would really have helped to have a more proper set of results and ablations in Sec. 3 and establish the methodology of the paper on firmer grounds.
    - The motivation for the method in the introduction is that in any real-image dataset, one is not going to have access to all of the generative factors of variation, and thus, one would want to be robust to having only a subset of them being available. In what way do the real-world image experiments presented in the paper address this concern? How would [14] applied to the same task, with the same decoder architectures etc. perform on image manipulation?

**References**
[A]: https://en.wikipedia.org/wiki/Differential_entropy


**Time Spent Reviewing:**

16

---

> ### Author Response · Authors · 2021-08-10
> **Response to Reviewer Z3et**
>
> We thank the reviewer for such a detailed and dedicated review. We are glad that the reviewer finds the question asked by our work “very exciting”, and believe that the major concerns can be properly addressed in the scope of this rebuttal. We hope that given the amount of time that the reviewer has already invested in the review, the reviewer would be willing to re-assess the merits of our paper with the following clarifications in mind.
> ___
>
> **"There are a number of half-baked technical claims which are either incorrect or unclear (the distinction is not obvious to me in all the cases). For example, the paper claims that one can restrict the information in the labels by maximizing the entropy of the label distribution from the attribute classifier given an image. I imagine one wants to stop the information from the image from leaking into the classifier? If so, denoting by Y the attribute labels and X the image, the (Shannon) mutual information between X and Y is H(X) - H(Y | X). Thus, if one wanted the labels Y to have low information about the image, one would actually maximize the conditional entropy not minimize it! (*)"**: We believe this criticism is erroneous and is based on several technical mistakes. First of all, the mutual information between two variables $X$ and $Y$ is $I(X,Y) = H(X) - H(X | Y) = H(Y) - H(Y | X)$ and not as stated by the reviewer. More importantly, we do not optimize the mutual information!
> We simply optimize the prior that the labels of each example can take a single value rather than a general distribution on the possible values of $y$. Additionally, restricting the range of classifier outputs, limits its expressivity, therefore encouraging the attribute value of each example to be close to the one hot vectors describing the known attributes. Otherwise the encoder could easily leak knowledge about the residual attributes. We therefore insist that our claims are correct. We will add this clarification to the paper.
>
> **"Another claim is that the information in a Gaussian random variable can be minimized by regularizing the mean to have a low L2-norm. Even if one just looked at the entropy of a Gaussian (and not the conditional entropy) and assumed that someone minimizing it was the right thing to do (which the previous bullet point indicates might not be), the differential entropy of a Gaussian [A] actually does not even depend on the mean, and only depends on the variance. Thus, in some sense the norm of the means has nothing to do with the information in the Gaussian distribution. (*)"**: We believe that the reviewer might have misunderstood the regularization of the residual attributes. This regularization form was properly justified in [14]. We will briefly explain it here (and in the final version of the paper for completeness). As the residual attributes are completely unsupervised, we regularize the respective latent variables, similarly to variational auto-encoders (VAEs) i.e. matching their distribution to a prior Normal distribution (with zero mean, and unit covariance). The term simply translates to:
> \begin{equation}
> D_{KL}(\mathcal{N}(\mu, \sigma^2 I) || \mathcal{N}(0, I)) = -\frac{1}{2} \sum_d (\ln{\sigma_d^2}-\sigma_d^2-\mu_d^2 + 1)
> \end{equation}
> Where $\mu$ and $\sigma$ vectors express the mean and standard deviation of the posterior latent distribution of the residual attributes (given an input image $x$). According to a key finding presented in [14], we do not learn the variance of the posterior distribution but rather set it fixed to  $\sigma_d^2=1$. The regularization term therefore translates to minimizing $||\mu||^2$, as stated in Eq. (6). To summarize, differently from the reviewer's interpretation, we do not regularize the mean of the group of all latent variables. Instead, we regularize each residual latent code $r$. Similarly to VAE, this sets a bottleneck on the amount of information in the latent code.
>
> **"It would have been interesting to study if one could disentangle some attributes given supervision of some other attributes to uncover the mechanisms of the world"**: We agree that reasoning the generative modeling based on a set of observed attributes for disentangling the rest of the attributes could open new exciting avenues for disentanglement research. Nonetheless, several non-trivial assumptions must hold for making a valid reasoning in real-world data (i.e. assuming that different factors of variations share similar mechanisms, which is obviously not always the case e.g. shape and texture). We believe that the preliminary question studied in our work is of great interest to the community (as other reviewers also agree). We hope that your suggestion does not detract from the legitimacy of our proposed setting, method and results.
>
> **"The general idea of the paper is very similar to that of [14] ... it is unclear what the novelty of the proposed approach is ...
> It would really help to have a concrete discussion of how the proposed approach and task are different from [14] in the related work section. ... what is the step that contributes to the gains of the proposed method?"**: As stated in the paper, we aim to achieve disentanglement in the *absence of full supervision on the attributes of interest (i.e. the supervised class in [14])*. We stress and show along the paper that methods as LORD [14] that indeed aim to disentangle the attributes of interest from a unified set of residual attributes, only work when full supervision is available on the attributes of
> interest and struggle when only partial labels are provided (see Tab. 1 in the paper and Fig. 5-7 in the appendix). More specifically, our method can be seen as an extension of LORD [14] for cases where the attributes of interest are observed only in a very few samples. Our method can therefore also leverage off-the-shelf zero-shot image classifiers such as CLIP, in order to be applied without the need to manually annotate even a small set of images. The few labels obtained with CLIP can not be used effectively by LORD [14], as demonstrated in our experiments.
>
> From a technical perspective, there are two fundamental differences between our method and LORD [14]: (i) LORD is a fully latent-based model i.e. no classifiers are trained with the generator in the first stage. The latent codes of the attributes of interest are optimized directly (and shared between all instances with the same label). Here we provide a hybrid latent-amortized approach where attribute codes are learned in a latent fashion, similarly to LORD, but they are weighted using the probabilities emitted by an amortized classifier.  (ii) We introduce an additional term $\mathcal{L}_{ent}$ which enables our method to perform well when very limited supervision exists for the attributes of interest.
>
> We show that our method strongly outperforms [14] in the few-label setting, achieving better quantitative scores and fundamentally more disentangled qualitative manipulations on real images (please see Fig. 5-7 in the appendix for a qualitative comparison with [14]).
>
> **"Would be nice to show / indicate the inference procedure in Fig. 1."**: Thank you for your suggestion, we will visualize the inference procedure in the sketch for completeness.
>
> **"In general, it would be helpful to see an ablation study of the contribution of each of the terms in Eqn. (8) to the performance."**: We provide an ablation study of the different terms in Eq. (8), as per your request.
>
> - ***Ablation for $\mathcal{L}_{ent}$ on Shapes3d, using 1000 [or 100] labels per attribute of interest***
>
> |                              |      D      |      C      |      I      |     SAP     |     MIG     |
> |------------------------------|:-----------:|:-----------:|:-----------:|:-----------:|:-----------:|
> | Ours w/o $\mathcal{L}_{ent}$ | 0.99 [0.99] | 0.98 [0.98] | 0.98 [0.98] | 0.28 [0.26] | 0.94 [0.91] |
> | Ours                         | 1.0 [1.0]   | 1.0 [1.0]   | 1.0 [1.0]   | 0.30 [0.30] | 1.0 [0.96]  |
>
> - ***Ablation for $\mathcal{L}_{ent}$ on Cars3d, using 1000 [or 100] labels per attribute of interest***
>
> |                              |       D      |      C      |      I      |     SAP     | MIG         |
> |----------------------------|:------------:|:-----------:|:-----------:|:-----------:|-------------|
> | Ours w/o $\mathcal{L}_{ent}$ | 0.74 [0.39]  | 0.74 [0.40] | 0.71 [0.43] | 0.22 [0.11] | 0.57 [0.34] |
> | Ours                         | 0.80 [0.40]  | 0.80 [0.41] | 0.78 [0.56] | 0.33 [0.15] | 0.61 [0.35] |
>
> - ***Ablation for $\mathcal{L}_{res}$ on Shapes3d, using 1000 labels per attribute of interest*** (accuracy of classifying the attributes of interest from the residual representations - lower is better)
>
> |                               | floor_color | wall_color | object_color |
> |-----------------------------|:----------:|:---------:|:-----------:|
> | Ours w/o $\mathcal{L}_{res}$  | 0.23       | 0.28      | 0.18        |
> | Ours                          | 0.11       | 0.12      | 0.14        |
> | Random Chance (optimal)       | 0.1        | 0.1       | 0.1         |
>
> The entire ablation study would be added to the final version.
>
> **"Is the evaluation only done on the specified attribute subset or for all the factors of variation in the dataset? Please clarify."**: We evaluate DCI, SAP and MIG only on the attributes of interest (Tab. 1 in the paper). Recall that we do not disentangle the residual attributes internally as no labels are provided at all. Tab. 3 in appendix measures the disentanglement of the residual attributes (as a unified set) from the attributes of interest.
>
> Please see next response for the remaining comments.

---

> > ### Author Response · Authors · 2021-08-10
> > **Response to Reviewer Z3et - Part 2**
> >
> > **"Table 2. It is not clear / not described what the pseudo-labels method here is."**: Section 3.4 describes the straightforward baseline of pseudo-labels. In this baseline, we explore whether we can obtain good performance using the simplest idea of training a classifier for each of the attributes of interest solely based on the provided few labels (i.e. 100 or 1000 labels). We show in Tab. 2 in the paper that training the classifiers together with the generator, as done in our method, improves the attribute classification over these straightforward pretrained attribute-wise classifiers.
> >
> > **"In general, it is very unclear how Sec. 4 results fit into the rest of the paper. While it is great that real image results are presented, it would really have helped to have a more proper set of results and ablations in Sec. 3 and establish the methodology of the paper on firmer grounds."**: As the setting proposed in this paper is highly-motivated by the challenge of attribute disentanglement in real images, where labels are often not available or require costly annotation, we believe these additional experiments strongly support the claims put forth along the paper. Nonetheless, we believe the additional results and ablations that we provided in this rebuttal, further strengthen the evaluation of the underlying approach.
> >
> > **"The motivation for the method in the introduction is that in any real-image dataset, one is not going to have access to all of the generative factors of variation. In what way do the real-world image experiments presented in the paper address this concern?"**: The experiments on real-world images rely on attribute annotations obtained with CLIP, for only some the attributes that describe images from the respective domains. For example, in the experiment on human faces (FFHQ), we use CLIP to partially annotate attributes as age, gender, hair color etc, while attributes as head pose, hair style, nose shape or identity-related attributes remain unknown. Similarly, in the experiment on animal faces (AFHQ), we partially annotate the species but have no supervision for the animal pose and background.
> >
> > **"How would [14] applied to the same task ..."**: The comparison with LORD [14] on real images already appears in the paper. Please see Fig. 5-7 in appendix for the comparison between the two methods (using the same network architecture).

---

> > ### Comment · Reviewer_Z3et · 2021-08-25
> > **Thank you for your response, updating my score.**
> >
> > Thanks to the authors for a detailed and thoughtful rebuttal. First of all, thanks for the clarification on first two points – that is on me, my concerns are resolved. Regarding third point "It would have been interesting to study if one could disentangle some attributes given supervision of some other attributes to uncover the mechanisms of the world" – the rebuttal is persuasive and well-taken, and I should clarify that I do not view it as a _deal-breaker_ – it does not weaken the contributions of this work. The ablation results look great – thank you for agreeing to add them in the final version. Overall, I am fairly convinced by the rebuttal and will update my score to 6.

---

> > > ### Author Response · Authors · 2021-08-26
> > > **Thank you for the updated score!**
> > >
> > > We appreciate your response and the time invested in the review. Thank you for acknowledging our clarifications and for the positive reconsideration of our work.

---

### Decision · Program_Chairs · 2021-09-27

**Decision:**

Accept (Poster)

**Comment:**

The problem setup with partially labelled images in practically meaningful. The whole paper is well organized and presented. The proposed disentangling method is well motivated and interesting.  The empirical results on a number of tasks are good. I really appreciate the detailed and informative rebuttals from the authors. It helped resolving the main concerns in the initial reviews.  All reviewers finally agree the acceptance.